# DATA-EFFICIENT HINDSIGHT OFF-POLICY OPTION LEARNING

## ABSTRACT

Hierarchical approaches for reinforcement learning aim to improve data efficiency and accelerate learning by incorporating different abstractions. We introduce Hindsight Off-policy Options (HO2), an efficient off-policy option learning algorithm, and isolate the impact of temporal and action abstraction in the option framework by comparing flat policies, mixture policies without temporal abstraction, and finally option policies; all with comparable policy optimization. We demonstrate the importance of off-policy optimization, as even flat policies trained off-policy can outperform on-policy option methods. In addition, off-policy training and backpropagation through a dynamic programming inference procedure – through time and through the policy components for every time-step – enable us to train all components' parameters independently of the data-generating behavior policy. Experimentally, we demonstrate that HO2 outperforms existing option learning methods and that both action and temporal abstraction provide strong benefits, particularly in more demanding simulated robot manipulation tasks from raw pixel inputs. We additionally illustrate challenges in off-policy option learning and highlight the importance of trust-region constraints. Finally, we develop an intuitive extension to further encourage temporal abstraction and investigate differences in its impact between learning from scratch and using pre-trained options.

## 1 INTRODUCTION

Despite deep reinforcement learning's considerable successes in recent years (Silver et al., 2017; OpenAI et al., 2018; Vinyals et al., 2019), applications in domains with limited or expensive data have so far been rare. To address this challenge, data efficiency can be improved through additional structure imposed on the solution space. Hierarchical methods, such as the options framework (Sutton et al., 1999; Precup, 2000), present one approach to integrate different abstractions into the agent. By representing an agent as a combination of low-level and high-level controllers, option policies can support reuse of low-level behaviours and can ultimately accelerate learning. The advantages introduced by the hierarchical control scheme imposed by the options framework are partially balanced by additional complexities, including possible degenerate cases (Precup, 2000; Harb et al., 2018), trade-offs regarding option length (Harutyunyan et al., 2019), and additional stochasticity. Overall, the interaction of algorithm and environment, as exemplified by the factors above, can become increasingly difficult, especially in an off-policy setting (Precup et al., 2006).

With Hindsight Off-policy Options (HO2), we present a method for data-efficient, robust off-policy learning of options to partially combat the previously mentioned challenges in option learning. We evaluate against current methods for option learning to demonstrate the importance of off-policy learning for data-efficiency. In addition, we compare HO2 with comparable policy optimization methods for flat policies and mixture policies without temporal abstraction. This allows us to isolate the individual impact of action abstraction (in mixture and option policies) and temporal abstraction in the option framework. The algorithm updates the policy via critic-weighted maximum-likelihood (similar to Abdolmaleki et al. (2018b); Wulfmeier et al. (2020)) and combines these with an efficient dynamic programming procedure to infer option probabilities along trajectories and update all policy parts via backpropagation through the inference graph (conceptually related to (Rabiner, 1989; Shiarlis et al., 2018; Smith et al., 2018)). Intuitively, the approach can be understood as inferring option and action probabilities for off-policy trajectories in hindsight and maximizing the likelihood of good actions and options by backpropagating through the inference procedure. To stabilize

policy updates, HO2 uses adaptive trust-region constraints, demonstrating the importance of robust policy optimization for hierarchical reinforcement learning (HRL) in line with recent work (Zhang & Whiteson, 2019). Rather than conditioning on executed options, the algorithm treats these as unobserved variables, computes the marginalized likelihood, and enables exact gradient computation and renders the algorithm independent of further approximations such as Monte Carlo estimates or continuous relaxation (Li et al., 2019). As an additional benefit, the formulation of the inference graph also allows to impose hard constraints on the option termination frequency, thereby regularizing the learned solution without introducing additional weighted loss terms that are dependent on reward scale. We exploit this to further investigate temporal abstraction for learning from scratch and for pre-trained options. Experimentally, we demonstrate that HO2 outperforms existing option learning methods and that both action and temporal abstraction provide strong benefits, particularly in more demanding simulated robot manipulation tasks from raw pixel inputs.

Our main contributions include:

- a data-efficient off-policy option learning algorithm which outperforms existing option learning methods on common benchmarks.

- careful analysis to isolate the impact of action abstraction and temporal abstraction in the option framework by comparing flat policies, mixture policies without temporal abstraction, and finally option policies. Our experiments demonstrate individual improvements from both forms of abstraction in complex 3D robot manipulation tasks from raw pixel inputs, and include ablations of other factors such as trust region constraints and off-policy versus on-policy data.

- an intuitive method to further encourage temporal abstraction beyond the core method, using the inference graph to constrain option switches without additional weighted loss terms. We investigate differences between temporal abstraction in learning from scratch and pre-trained options and show that optimizing for temporally abstract behaviour in addition to simply providing the methods for its emergence mostly provides benefits in the context of pre-trained options.

## 2 PRELIMINARIES

**Problem Setup**    We consider a reinforcement learning setting with an agent operating in a Markov Decision Process (MDP) consisting of the state space $\mathcal{S}$, the action space $\mathcal{A}$, and the transition probability $p(s_{t+1}|s_t, a_t)$ of reaching state $s_{t+1}$ from state $s_t$ when executing action $a_t$. The actions $a_t$ are drawn from the agent's policy $\pi(a_t|x_t)$, where $x_t$ can either refer to the current state $s_t$ of the agent or, in order to model dependencies on the previous steps, the trajectory until the current step, $h_t = \{s_t, a_{t-1}, s_{t-1}, ...s_0, a_0\}$. Jointly, the transition dynamics and policy induce the marginal state visitation distribution $p(s_t)$. The discount factor $\gamma$ together with the reward $r_t = r(s_t, a_t)$ gives rise to the expected return, which the agent aims to maximize: $J(\pi) = \mathbb{E}_{p(s_t), \pi(a_t, s_t)}\left[\sum_{t=0}^{\infty} \gamma^t r_t\right]$.

We start by describing mixture policies as an intermediate between flat policies and the option framework (see Figure 1), which introduce a type of action abstraction via multiple low-level policies. Note that both Gaussian and mixture policies have been trained in prior work via similar policy optimization methods (Abdolmaleki et al., 2018a; Wulfmeier et al., 2020), which we will extend towards option policies. In particular, we will make use of the connection to isolate the impact of action abstraction and temporal abstraction in the option framework in Section 4.2. The next paragraphs focus on computing likelihoods of actions and options, which forms the foundation for the proposed critic-weighted maximum likelihood algorithm to learn hierarchical policies.

**Mixture Policies**    can be seen as a simplification of the options framework without initiation or termination condition, with resampling of options after every step (i.e. no dependency between the options of timestep $t$ and $t+1$ in Figure 1). The joint probability of action and option is given as:

$$\pi_\theta(a_t, o_t|s_t) = \pi^L\left(a_t|s_t, o_t\right)\pi^H\left(o_t|s_t\right), \ \text{ with } \pi^H\left(o_t|s_t\right) = \pi^C\left(o_t|s_t\right), \quad (1)$$

where $\pi^H$ and $\pi^L$ respectively represent high-level policy (which for the mixture is equal to a Categorical distribution $\pi^C$) and low-level policy (components of the resulting mixture distribution), and $o$ is the index of the sub-policy or mixture component.

## 3 METHOD

**Option Policies** extend mixture policies by incorporating temporal abstraction. We follow the semi-MDP and *call-and-return* option model (Sutton et al., 1999), defining an option as a triple $(I(s_t, o_t), \pi^L(a_t|s_t, o_t), \beta(s_t, o_t))$. The initiation condition $I$ describes an option's probability to start in a state and is adapted as $I(s_t, o_t) = 1 \forall s_t \in \mathcal{S}$ following (Bacon et al., 2017; Zhang & Whiteson, 2019). The termination condition $b_t \sim \beta(s_t, o_t)$ denotes a Bernoulli distribution describing the option's probability to terminate in any given state, and the action distribution for a given option is modelled by $\pi^L(a_t|s_t, o_t)$. Every time the agent observes a state, the current option's termination condition is sampled. If subsequently no option is active, a new option is sampled from the controller $\pi^C(o_t|s_t)$. Finally, we sample from either the continued or new option to generate a new action.

The transitions between options can therefore be described by

$$p(o_t|s_t, o_{t-1}) = \begin{cases} 1 - \beta(s_t, o_{t-1}) + \beta(s_t, o_{t-1})\pi^C(o_t|s_t) & \text{if } o_t = o_{t-1} \\ \beta(s_t, o_{t-1})\pi^C(o_t|s_t) & \text{otherwise} \end{cases} \quad (2)$$

Equation 3 describes the direct connection between mixture and option policies. In both cases, the low-level policies $\pi^L$ only depend on the current state. However, where mixtures only depend on the current state $s_t$ for the high-level probabilities $\pi^H$, options take into account the history $h_t$ directly via the previous option as described in Equation 4.

$$\pi_\theta(a_t, o_t|h_t) = \pi^L(a_t|s_t, o_t)\,\pi^H(o_t|h_t) \quad (3)$$

Following the graphical model in Figure 1, the probability of being in component $o_t$ at timestep $t$ across a trajectory $h_t$ can be determined in a recursive manner based on the option probabilities of the previous timestep. For the first timestep, the probabilities are given by the high-level controller $\pi^H(o_0|h_0) = \pi^C(o_0|s_0)$ and for all consecutive steps are computed as follows for $M$ options using the option transition probabilities in Equation 2.

The factorization in Equation 3 and the following exact per-timestep marginalization over options in Equation 4 enable us to efficiently compute the likelihood of actions and options along off-policy trajectories (instead of using option samples) to efficiently perform intra-option learning (Precup, 2000) for all options independently of the executed option.

$$\tilde{\pi}^H(o_t|h_t) = \sum_{o_{t-1}=1}^{M} \left[ p(o_t|s_t, o_{t-1})\,\pi^H(o_{t-1}|h_{t-1})\,\pi^L(a_{t-1}|s_{t-1}, o_{t-1}) \right] \quad (4)$$

In every step, we normalize the distribution following $\pi^H(o_t|h_t) = \tilde{\pi}^H(o_t|h_t)/\sum_{o'_t=1}^{M} \tilde{\pi}^H(o'_t|h_t)$.

This dynamic programming formulation (similar to (Rabiner, 1989; Shiarlis et al., 2018; Smith et al., 2018)) allows the use of automatic differentiation in modern deep learning frameworks (e.g. (Abadi et al., 2016)) to backpropagate through the graph to determine the gradient updates for all policy parameters during the policy improvement step. We compute the probability of the option $o_t$ at time $t$ given past states and actions by marginalizing out the previous option. Since $o_{t-1}$ affects $a_{t-1}$, we need to take the previous action that was actually chosen into account when performing the marginalization. The method is computationally more efficient than determining updates over all possible sequences of options independently and reduces variance compared to sampling-based approximations with the executed option or new samples during learning. In practice we find that

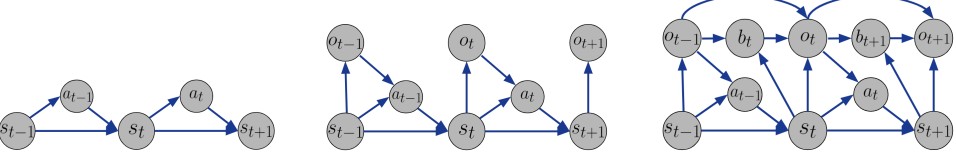

Figure 1: Graphical model for flat policies (left), mixture policies (middle) - introducing a type of action abstraction, and option policies (right) - adding temporal abstraction via autoregressive options. While the action $a$ is solely dependent on the state $s$ for flat policies, mixture policies introduce the additional component or option $o$ which affects the actions (following Equation 1). Option policies do not change the direct dependencies for actions but instead affect the options themselves, which are now also dependent on the previous option and its potential termination $b$ (following Equation 2).

Figure 2: Representation of the dynamic programming forward pass - bold arrows represent connections without switching. Left: example with two options. Right: extension of the graph to explicitly count the number of switches. Marginalization over the dimension of switches determines component probabilities. By limiting over which nodes to sum at every timestep, the optimization can be targeted to fewer switches and more consistent option execution.

removing the conditioning of component probabilities on actions (the $\pi_L$ terms in Equation 4) improves performance and stability. This is furthermore shown in the ablations for the effects of conditioning on potentially stale actions in off-policy learning in Section 4.3.

We now continue by describing the main policy improvement algorithm, which requires the previously determined option probabilities. The principal three steps are: 1. update the critic following Eq. 5; 2. generate an intermediate, non-parametric policy based on the updated critic (Eq. 6); 3. update the parametric policy to align to the non-parametric improvement (Eq. 8).

**Policy Evaluation** In comparison to training mixture policies (Wulfmeier et al., 2020), the critic for option policies is a function of $s$, $a$, and $o$ since the current option influences the likelihood of future actions and thus rewards. Note that even though we express the policy as a function of the history $h_t$, $Q$ is a function of $o_t, s_t, a_t$, since these are sufficient to render the future trajectory independent of the past (see the graphical model in Figure 1). In the constrained case $Q$ also depends on $n_t$, which we omit here for clarity. We define the TD(0) objective as

$$\min_\phi L(\phi) = \mathbb{E}_{s_t, a_t, o_t \sim \mathcal{D}} \Big[ (Q_{\mathrm{T}} - Q_\phi(s_t, a_t, o_t))^2 \Big], \tag{5}$$

where the current states, actions, and options are sampled from the current replay buffer $\mathcal{D}$. For the 1-step target $Q_{\mathrm{T}} = r_t + \gamma \mathbb{E}_{s_{t+1}, a_{t+1}, o_{t+1}}[Q'(s_{t+1}, a_{t+1}, o_{t+1})]$, the expectation over the next state is approximated with the sample $s_{t+1}$ from the replay buffer, and we estimate the value by sampling actions and options according to $a_{t+1}, o_{t+1} \sim \pi^H(\cdot|h_{t+1})$ following Equation 2.

**Policy Improvement** We follow an Expectation-Maximization procedure similar to (Wulfmeier et al., 2020; Abdolmaleki et al., 2018b), which first computes an improved non-parametric policy and then updates the parametric policy to match this target. Given $Q$, all we require to optimize option policies is the possibility to sample from the policy and determine the log-likelihood (gradient) of actions and options under the policy given $h_t$. The first step of policy improvement provides us with the non-parametric policy $q(a_t, o_t|h_t)$.

$$\max_q J(q) = \mathbb{E}_{a_t, o_t \sim q, s_t \sim \mathcal{D}} \big[ Q_\phi(s_t, a_t, o_t) \big], \ \text{ s.t. } \mathbb{E}_{h_t \sim \mathcal{D}} \Big[ \mathrm{KL}\big( q(\cdot|h_t) \| \pi_\theta(\cdot|h_t) \big) \Big] \le \epsilon_E, \tag{6}$$

where $\mathrm{KL}(\cdot\|\cdot)$ denotes the Kullback-Leibler divergence, and $\epsilon_E$ defines a bound on the KL. We can find the solution to the constrained optimization problem in Equation 6 in closed-form and obtain

$$q(a_t, o_t|h_t) \propto \pi_\theta(a_t, o_t|h_t) \exp \left( Q_\phi(s_t, a_t, o_t)/\eta \right). \tag{7}$$

Practically speaking, this step computes samples from the previous policy and weights them based on the corresponding temperature-calibrated values of the critic. The temperature parameter $\eta$ is computed following the dual of the Lagrangian. The derivation and final form of the dual can be found in Appendix C.1, Equation 17.

To align parametric and non-parametric policies in the second step, we minimize their KL divergence.

$$\theta_+ = \arg\min_\theta \mathbb{E}_{h_t \sim \mathcal{D}} \Big[ \mathrm{KL}\big( q(\cdot|h_t) \| \pi_\theta(\cdot|h_t) \big) \Big], \text{s.t. } \mathbb{E}_{h_t \sim \mathcal{D}} \Big[ \mathcal{T}\big( \pi_{\theta_+}(\cdot|h_t) \| \pi_\theta(\cdot|h_t) \big) \Big] \le \epsilon_M \tag{8}$$

The distance function $\mathcal{T}$ in Equation 8 has a trust-region effect and stabilizes learning by constraining the change in the parametric policy. The computed option probabilities from Equation 4 are used in Equation 7 to enable sampling of options as well as Equation 8 to determine and maximize the likelihood of option samples under the policy. We can apply Lagrangian relaxation again and solve the primal as detailed in Appendix C.2. Finally, we describe the complete pseudo-code for HO2 in Algorithm 1, where we use target networks for policy $\pi'$ and Q-function $Q'$ to stabilize training.

---

**Algorithm 1** Hindsight Off-policy Options

---

**input:** initial parameters for $\theta$, $\eta$ and $\phi$, KL regularization parameters $\epsilon$, set of trajectories $\tau$
**while** not done **do**
    sample trajectories $\tau$ from replay buffer
    // forward pass along sampled trajectories
    determine component probabilities $\pi^H(o_t|h_t)$ (Eq. 4)
    sample actions $a_j$ and options $o_j$ from $\pi_\theta(\cdot|h_t)$ (Eq. 3) to estimate expectations
    // compute gradients over batch for policy, Lagrangian multipliers and Q-function
    $\delta_\theta \leftarrow -\nabla_\theta \sum_{h_t \in \tau} \sum_j [\exp(Q_\phi(s_t, a_j, o_j)/\eta) \log \pi_\theta(a_j, o_j|h_t)]$ following Eq. 7 and 8
    $\delta_\eta \leftarrow \nabla_\eta g(\eta) = \nabla_\eta \eta \epsilon + \eta \sum_{h_t \in \tau} \log \sum_j [\exp(Q_\phi(s_t, a_j, o_j)/\eta)]$ following Eq. 17
    $\delta_\phi \leftarrow \nabla_\phi \sum_{(s_t, a_t, o_t) \in \tau} \left(Q_\phi(s_t, a_t, o_t) - Q_T\right)^2$ following Eq. 5
    update $\theta, \eta, \phi$    // apply gradient updates
    **if** number of iterations = target update **then**
        $\pi' = \pi_\theta, Q' = Q_\phi$    // update target networks for policy $\pi'$ and value function $Q'$

---

**Temporal Consistency**    Using options over longer sequences can help to reduce the search space and simplify exploration (Sutton et al., 1999; Harb et al., 2018). Previous approaches (e.g. (Harb et al., 2018)) rely on additional weighted loss terms which penalize option transitions. We instead introduce a mechanism to explicitly limit the maximum number of switches between options along a trajectory to increase temporal consistency. In comparison to additional loss terms, a parameter for the maximum number of switches can be chosen independently of the reward scale of an environment and provides an intuitive semantic interpretation. Both aspects simplify hyperparameter tuning.

The 2D graph for computing option probabilities along a trajectory in Figure 2 is extended with a third dimension representing the number of switches between options. Practically, this means that we are modelling $\pi^H(o_t, n_t|h_t)$ where $n_t$ represents the number of switches until timestep $t$. We can marginalize over all numbers of switches to determine option probabilities. In order to encourage option consistency across timesteps, we can instead sum over only a subset of nodes for all $n \leq N$ with $N$ smaller than the maximal number of switches leading to $\pi^H(o_t|h_t) = \sum_{n_t=0}^N \pi^H(o_t, n_t|h_t)$.

For the first timestep, only 0 switches are possible and $\pi^H(o_0, n_0|h_0) = 0$ everywhere except for $n_0 = 0$ and $\pi^H(o_0|n_0 = 0, h_0) = \pi^C(o_0|s_0)$. For further timesteps, all edges following option terminations $\beta$ lead to the next step's option probabilities with increased number of switches $n_{t+1} = n_t + 1$. All edges following option continuation lead to the probabilities for equal number of switches $n_{t+1} = n_t$. This results in the computation of the joint distribution for $t > 0$ in Equation 9 with the option and switch index transitions $p(o_t, n_t|s_t, o_{t-1}, n_{t-1})$ described in Equation 19 in the Appendix and normalization as $\pi^H(o_t, n_t|h_t) = \tilde{\pi}^H(o_t, n_t|h_t)/\sum_{o'_t=1}^M \sum_{n'_t=1}^L \tilde{\pi}^H(o'_t, n'_t|h_t)$.

$$\tilde{\pi}^H(o_t, n_t|h_t) = \sum_{\substack{o_{t-1}=1, \\ n_{t-1}=1}}^{M,N} p(o_t, n_t|s_t, o_{t-1}, n_{t-1}) \pi^H(o_{t-1}, n_{t-1}|h_{t-1}) \pi^L(a_{t-1}|s_{t-1}, o_{t-1}) \quad (9)$$

## 4 EXPERIMENTS

In this section, we aim to answer a set of questions to better understand the contribution of different aspects to the performance of option learning - in particular with respect to the proposed method, HO2. Across domains, we use MPO (Abdolmaleki et al., 2018a) to train flat Gaussian policies, RHPO (Wulfmeier et al., 2020) for mixture-of-Gaussians policies as an intermediate step, and HO2 to train option policies - all based on equivalent policy optimization (types of critic-weighted maximum likelihood estimation).

To start, in Section 4.1 we explore (1) How well does HO2 perform in comparison to existing option learning methods? and (2) How important is off-policy training in this context? We use a set of common OpenAI gym (Brockman et al., 2016) benchmarks to answer these questions (Section 4.1). In Section 4.2 we ask (3) How do action abstraction in mixture policies and the additional temporal abstraction brought by option policies individually impact performance? We use more complex, pixel-based 3D robotic manipulation experiments to investigate these two aspects and

Figure 3: Results on OpenAI gym. Dashed black line represents DAC (Zhang & Whiteson, 2019), dotted line represents Option-Critic (Bacon et al., 2017) and solid line represents IOPG (Smith et al., 2018) (lines reflect the approximate results after $2 \times 10^6$ steps from (Zhang & Whiteson, 2019)). We limit the number of switches to 5 for HO2-limits. HO2 outperforms existing option learning algorithms.

evaluate scalability with respect to higher dimensional input and state spaces. In Section 4.2.1, we explore (4) How does increased temporal consistency impact performance in particular with respect to sequential transfer with pre-trained options (applying the constrained formulation from Section 3)?. Finally, we perform additional ablations in Section 4.3 to investigate the challenges of robust off-policy option learning and improve understanding of various environment and algorithm aspects.

## 4.1 COMPARISON OF OPTION LEARNING METHODS

We compare HO2 (with and without limiting option switches) against competitive baselines for option learning in common, feature-based continuous action space domains. The baselines include Double Actor-Critic (DAC) (Zhang & Whiteson, 2019), Inferred Option Policy Gradient (IOPG) (Smith et al., 2018) and Option-Critic (OC) (Bacon et al., 2017).

As demonstrated in Figure 3, off-policy learning alone (here MPO (Abdolmaleki et al., 2018b)) improves data-efficiency and can suffice to outperform on-policy option algorithms such as DAC, IOPG and Option-Critic, for example in the HalfCheetah and Swimmer domains while otherwise at least performing on par. This emphasizes the importance of a strong underlying policy optimization method. We achieve further improvements when training mixture policies via RHPO (Wulfmeier et al., 2020) and can accelerate learning with temporal abstraction via HO2. Overall, HO2performs better or commensurate to MPO and RHPO across all tasks and show strong gains in the HalfCheetah and Swimmer tasks. Using the switch constraints for increasing temporal abstraction from Section 3 has overall only a minor effect on performance. We further investigate this effect in sequential transfer in Section 4.2.1. Given the comparable simplicity of these tasks and correlated solutions, principal gains are associated with off-policy training. In the next section, we study more complex domains to isolate gains from action and temporal abstraction.

## 4.2 INVESTIGATING ACTION ABSTRACTION AND TEMPORAL ABSTRACTION

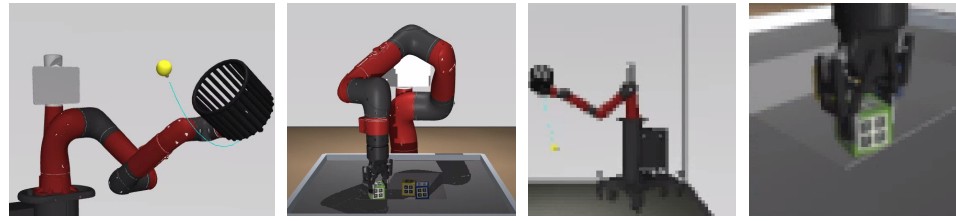

Figure 4: Ball-In-Cup and Stacking. Left: Environments. Right: Example agent observations.

We next investigate different aspects of option learning via HO2 on more complex simulated robotic manipulation tasks - stacking and ball-in-cup (BIC) - as displayed in Figure 4, based on robot proprioception and raw pixel inputs (64x64 pixel, 2 cameras for BIC and 3 for stacking). Since the performance of HO2 for training from scratch is relatively independent of switch constraints as seen in Figure 3 we will focus on the base method only. To reduce data requirements for these more complex tasks, we use a set of known methods to improve data-efficiency and accelerate learning for all methods. We will apply multi-task learning with a related set of tasks to provide a curriculum and share policies and Q-functions across tasks with details in Appendix A. Furthermore,

we assign rewards for all tasks to any generated transition data in hindsight to share data and simplify exploration (Andrychowicz et al., 2017; Riedmiller et al., 2018; Wulfmeier et al., 2020).

Across all tasks, except for simple positioning and reach tasks (see Appendix A), action abstraction improves performance (mixture policy versus flat Gaussian policies). In particular, the more challenging later stacking tasks shown in Figure 5 intuitively benefit from shared sub-behaviours with earlier tasks such as grasping and lifting, demonstrating accelerated learning. The introduction of temporal abstraction (option policy vs mixture policy) further accelerates learning. Similarly to action abstraction via mixtures (RHPO), the impact increases with task complexity and is particularly pronounced for the stacking tasks.

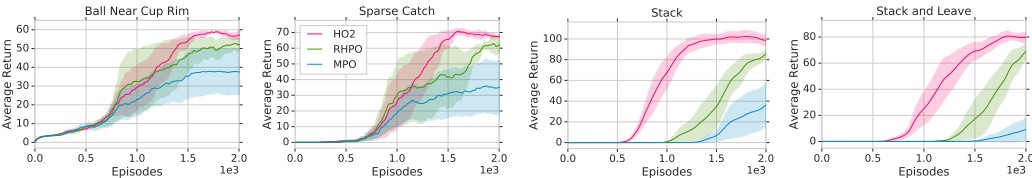

Figure 5: Results for option policies, mixture policies, and single Gaussian policies (respectively HO2, RHPO, and MPO) in multi-task domains with pixel-based ball-in-cup (left) and pixel-based block stacking (right). All four tasks displayed use sparse binary rewards, such that the obtained reward represents the number of timesteps where the corresponding condition - such as the ball is in the cup - is fulfilled. Please see Appendix B for further details and additional tasks.

### 4.2.1 OPTIMIZING FOR TEMPORAL ABSTRACTION

There is a difference between enabling the policy to represent temporal abstraction and explicitly maximizing it. The ability to represent temporal abstract behaviour in HO2 consistently helps in previous experiments. However, there is limited benefit in increasing temporal consistency (via limiting the number of switches) in our experiments for training from scratch in Section 4.1. In this section, we evaluate this factor for sequential transfer with pre-trained options. We first train low-level options for all tasks except for the final, most complex task in each domain. Successively, given this set of pre-trained options, we only train for the final task and compare training with and without limited switches. We build on the same domains from Section 4.2: block stacking and BIC based on the Sawyer robot. As shown in Figure 6, across both domains, we can see that more consistent options lead to increased performance in the transfer domain. Intuitively, increased temporal consistency and fewer switches lead to a smaller search space from the perspective of the high-level controller, by persisting with one behavior for longer after selection. While the same mechanism can also hold for training from scratch, it is likely that the added complexity of maximizing temporal consistency simultaneous to learning the low-level behaviours outweighs the benefits.

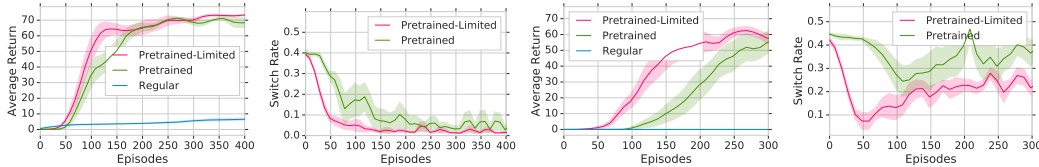

Figure 6: Sequential transfer experiments with limited option switches. Left: BIC. Right: Stack. We see considerable improvements for limited switches. In addition, we visualize the actual agent option switch rate in the environment to directly demonstrate the constraint's effect.

### 4.3 FURTHER ANALYSIS AND ABLATIONS

In this section, we investigate different algorithmic aspects to get a better understanding of the method, properties of the learned options, and how to achieve robust training in the off-policy setting.

**Off-policy Option Learning** Figure 7 visualizes that across most of our experiments, we find that conditioning component probabilities on the previous timesteps' action probabilities (see Section 3)

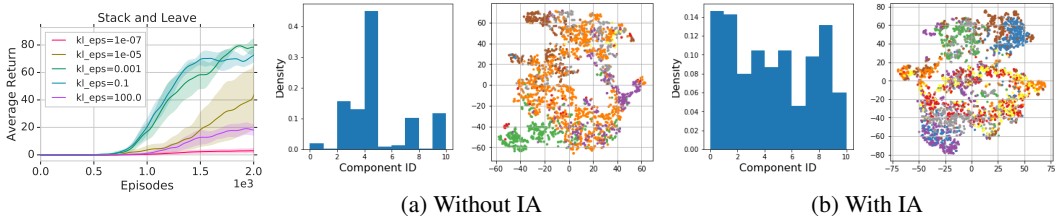

Figure 7: Results on OpenAI gym with/without conditioning of option probabilities on past actions.

detrimental. An explanation for these results can be found in the potential staleness of performed actions with respect to the current policy caused by the off-policy training setup. We additionally include results displaying less impact in the on-policy case in Appendix A.4 to demonstrate the connection between the effect and the off-policy setting. While we use a simple empirical change, the problem has lead to various off-policy corrections for goal-based HRL (Nachum et al., 2018b; Levy et al., 2017) which provide a valuable direction for future work.

**Trust-regions and Robustness**   Applying trust-region constraints for policy updates has been shown to be beneficial for non-hierarchical policies (Schulman et al., 2015; Abdolmaleki et al., 2018b). In this section, we describe the impact of ablating the strength of constraints on the option probability updates (both termination conditions $\beta$ and the high-level controller $\pi_C$). As displayed in Figure 8, the approach is robust across different values, but very weak or strong constraints can considerably degrade performance. Note that a high value is essentially equal to no constraint and causes very low performance. Training option policies strongly relies here on trust-region constraints during optimization. Details for additional tasks in the domain can be found in Appendix A.5.

Figure 8: Block stacking results with different trust-region constraints.

Figure 9: Analysis on Ant locomotion tasks, showing histogram over options, and t-SNE scatter plots in action space colored by option.

(a) Without IA  (b) With IA

**Decomposition and Option Clustering**   We apply HO2 to a variety of simple locomotion tasks to investigate how the agent uses its capacity and decomposes behavior into options. In these tasks, the agent body (eg. "Ant") must go to one of three targets in a room, with the task specified by the target locations and a selected target index. As shown in Figure 9, we find that option decompositions depend on both the task properties and algorithm settings. *Information asymmetry (IA)*, here achieved by providing task information only to the high-level controller, can address degenerate solutions and lead to increased diversity with respect to options (as shown by the histogram over options) and more specialized options (represented by the clearer clustering of samples in action space). More extensive experiments and discussion with a wider variety of bodies and tasks, with numerous quantitative and qualitative measures of option decomposition, can be found in Appendix A.1. To summarize, these analyses yield a number of additional interesting observations, showing that (1) for simpler bodies like a "Ball", the options are easily interpretable (forward torque, and turning left/right at different rates); and (2) applying the switch constraint introduced in Section 4.2.1 leads to a much lower switch rate while allowing the agent to solve the task.

## 5   RELATED WORK

Hierarchy has been investigated in many forms in reinforcement learning to improve data gathering as well as data fitting aspects. Goal-based approaches commonly define a grounded interface between high- and low-level policies such that the high level acts by providing goals to the low level, which is trained to achieve these goals (Dayan & Hinton, 1993; Levy et al., 2017; Nachum et al., 2018a;b;

Vezhnevets et al., 2017), effectively separating objectives and improving exploration. These methods have been able to overcome very sparse reward domains but commonly require domain knowledge and a crafted interface can limit expressiveness and richness of behaviour.

More emergent interfaces within policies have been investigated from an RL-as-inference perspective via policies with continuous latent variables (Haarnoja et al., 2018; Hausman et al., 2018; Heess et al., 2016; Igl et al., 2019; Tirumala et al., 2019; Teh et al., 2017). Related to these approaches, we provide a probabilistic inference perspective to off-policy option learning and benefit from efficient dynamic programming inference procedures. We furthermore build on the related idea of information asymmetry (Pinto et al., 2017; Galashov et al., 2018; Tirumala et al., 2019) - providing a part of the observations only to a part of the model. The asymmetry can lead to an information bottleneck affecting the properties of learned low-level policies. We build on the intuition and demonstrate how option diversity can be affected in ablations in Section 4.3.

Our work builds on and investigates the option framework (Precup, 2000; Sutton et al., 1999), which describes policies with an autoregressive, discrete latent space. Option policies commonly use a high-level controller to choose from a set of options or skills which additionally include termination conditions, to enable a skill to represent temporally extended behaviour. Without termination conditions, options can be seen as equivalent to components under a mixture distribution, and this simplified formulation has been applied successfully in different methods (Agostini & Celaya, 2010; Daniel et al., 2016; Wulfmeier et al., 2020). The option framework has been further extended and improved for more practical application (Bacon et al., 2017; Harb et al., 2018; Harutyunyan et al., 2019; Precup et al., 2006; Riemer et al., 2018; Smith et al., 2018). Finally, the benefits of options and other modular policy styles have also been applied in the supervised case for learning from demonstration (Fox et al., 2017; Krishnan et al., 2017; Shiarlis et al., 2018).

One important step towards increasing the robustness of option learning has been taken in (Zhang & Whiteson, 2019) by building on robust (on-policy) policy optimization with PPO (Schulman et al., 2017). HO2 has similar robustness benefits, but additionally improves data-efficiency by building on off-policy learning, hindsight inference of options, and additional trust-region constraints (Abdolmaleki et al., 2018b; Wulfmeier et al., 2020). Related inference procedures have also been investigated in imitation learning (Shiarlis et al., 2018) as well as on-policy RL (Smith et al., 2018). In addition to inference of suited options in hindsight, off-policy learning enables us to make use of hindsight assignment of rewards for multiple tasks, which has been successfully applied with flat, non-hierarchical policies (Andrychowicz et al., 2017; Riedmiller et al., 2018) and goal-based hierarchical approaches (Levy et al., 2017; Nachum et al., 2018b).

## 6 CONCLUSIONS

We introduce a novel actor-critic algorithm for robust and efficient off-policy training of option policies. The approach outperforms recent work in option learning on common benchmarks and is able to solve complex, simulated robot manipulation tasks from raw pixel inputs more reliably than competitive baselines. HO2 infers option and action probabilities for trajectories in hindsight, and performs critic-weighted maximum-likelihood estimation by backpropagating through the inference procedure. The ability to infer option choices given a trajectory, allows us to train from off-policy trajectories, including those from different tasks and makes it possible to impose hard constraints on the termination frequency without introducing additional weighted objectives.

We separately analyze the impact of action abstraction (via mixture policies), and temporal abstraction (via the autoregressive modeling of options), and find that each abstraction independently improves performance. The additional enforcing of temporal consistency for option choices is beneficial when transferring pre-trained options to a new task but displays a limited effect when learning from scratch. Furthermore, we investigate the consequences of the off-policyness of training data and demonstrate the benefits of trust-region constraints for option learning. Finally, we examine the impact of different agent and environment properties (such as information asymmetry, tasks, and embodiments) with respect to task decomposition and option clustering; which provides a relevant future direction. Finally, since our method performs (weighted) maximum likelihood learning, it can be adapted naturally to learn structured behavior representations in mixed data regimes, e.g. to learn from combinations of demonstrations, logged data, and online trajectories. This opens up promising directions for future work.

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

# A    ADDITIONAL EXPERIMENTS

## A.1    DECOMPOSITION AND OPTION CLUSTERING

We further deploy HO2 on a set of simple locomotion tasks, where the goal is for an agent to move to one of three randomized target locations in a square room. These are specified as a set of target locations and a task index to select the target of interest.

The main research questions we aim to answer (both qualitatively and quantitatively) are: (1) How do the discovered options specialize and represent different behaviors?; and (2) How is this decomposition affected by variations in the task, embodiment, or algorithmic properties of the agent? To answer these questions, we investigate a number of variations:

- Three bodies: a quadruped with two ("Ant") or three ("Quad") torque-controlled joints on each leg, and a rolling ball ("Ball") controlled by applying yaw and forward roll torques.
- With or without *information asymmetry* (IA) between high- and low-level controllers, where the task index and target positions are withheld from the options and only provided to the categorical option controller.
- With or without limited number of switches in the optimization.

Information-asymmetry (IA) in particular, has recently been shown to be effective for learning general skills: by withholding task-information from low-level options, they can learn task-agnostic, temporally-consistent behaviors that can be composed by the option controller to solve a task. This mirrors the setup in the aforementioned Sawyer tasks, where the task information is only fed to the high-level controller.

For each of the different cases, we qualitatively evaluate the trained agent over 100 episodes, and generate histograms over the different options used, and scatter plots to indicate how options cluster the state/action spaces and task information. We also present quantitative measures (over 5 seeds) to accompany these plots, in the form of (1) Silhouette score, a measure of clustering accuracy based on inter- and intra-cluster distances[1]; and (2) entropy over the option histogram, to quantify diversity. The quantitative results are shown in Table 1, and the qualitative plots are shown in Figure 10. Details and images of the environment are in Section B.4.

| | | Scenario | Option entropy | Switch rate | Cluster score (actions) | Cluster score (states) | Cluster score (tasks) |
|---|---|---|---|---|---|---|---|
| **Regular** | Ball | No IA | $2.105 \pm 0.074$ | $0.196 \pm 0.010$ | $-0.269 \pm 0.058$ | $-0.110 \pm 0.025$ | $-0.056 \pm 0.011$ |
| | | With IA | $2.123 \pm 0.066$ | $0.346 \pm 0.024$ | $-0.056 \pm 0.024$ | $-0.164 \pm 0.051$ | $-0.057 \pm 0.008$ |
| | Ant | No IA | $1.583 \pm 0.277$ | $0.268 \pm 0.043$ | $-0.148 \pm 0.034$ | $-0.182 \pm 0.068$ | $-0.075 \pm 0.011$ |
| | | With IA | $2.119 \pm 0.073$ | $0.303 \pm 0.019$ | $-0.053 \pm 0.021$ | $-0.066 \pm 0.024$ | $-0.052 \pm 0.006$ |
| | Quad | No IA | $1.792 \pm 0.127$ | $0.336 \pm 0.019$ | $-0.078 \pm 0.064$ | $-0.113 \pm 0.035$ | $-0.089 \pm 0.050$ |
| | | With IA | $2.210 \pm 0.037$ | $0.403 \pm 0.014$ | $0.029 \pm 0.029$ | $-0.040 \pm 0.003$ | $-0.047 \pm 0.006$ |
| **Limited Switches** | Ball | No IA | $1.804 \pm 0.214$ | $0.020 \pm 0.009$ | $-0.304 \pm 0.040$ | $-0.250 \pm 0.135$ | $-0.131 \pm 0.049$ |
| | | With IA | $2.233 \pm 0.027$ | $0.142 \pm 0.015$ | $-0.132 \pm 0.035$ | $-0.113 \pm 0.043$ | $-0.053 \pm 0.003$ |
| | Ant | No IA | $1.600 \pm 0.076$ | $0.073 \pm 0.014$ | $-0.124 \pm 0.017$ | $-0.155 \pm 0.067$ | $-0.084 \pm 0.034$ |
| | | With IA | $2.222 \pm 0.043$ | $0.141 \pm 0.015$ | $-0.052 \pm 0.011$ | $-0.054 \pm 0.014$ | $-0.050 \pm 0.007$ |
| | Quad | No IA | $1.549 \pm 0.293$ | $0.185 \pm 0.029$ | $-0.075 \pm 0.036$ | $-0.126 \pm 0.030$ | $-0.112 \pm 0.022$ |
| | | With IA | $2.231 \pm 0.042$ | $0.167 \pm 0.025$ | $-0.029 \pm 0.029$ | $-0.032 \pm 0.004$ | $-0.053 \pm 0.009$ |

Table 1: Quantitative results indicating the diversity of options used (entropy), and clustering accuracy in action and state spaces (silhouette score), with and without information asymmetry (IA), and with or without limited number of switches. Higher values indicate greater separability by option / component.

The results show a number of trends. Firstly, the usage of IA leads to a greater diversity of options used, across all bodies. Secondly, with IA, the options tend to lead to specialized actions, as demonstrated by the clearer option separation in action space. In the case of the $2D$ action space of the ball, the options correspond to turning left or right (y-axis) at different forward torques (x-axis). Thirdly, while the simple Ball can learn these high-level body-agnostic behaviors, the options for

---

[1]The silhouette score is a value in $[-1, 1]$ with higher values indicating cluster separability. We note that the values obtained in this setting do not correspond to high *absolute* separability, as multiple options can be used to model the same skill or behavior abstraction. We are instead interested in the *relative* clustering score for different scenarios.

more complex bodies have greater switch rates that suggest the learned behaviors may be related to lower-level motor behaviors over a shorter timescale. Lastly, limiting the number of switches during marginalization does indeed lead to a lower switch rate between options, without hampering the ability of the agent to complete the task.

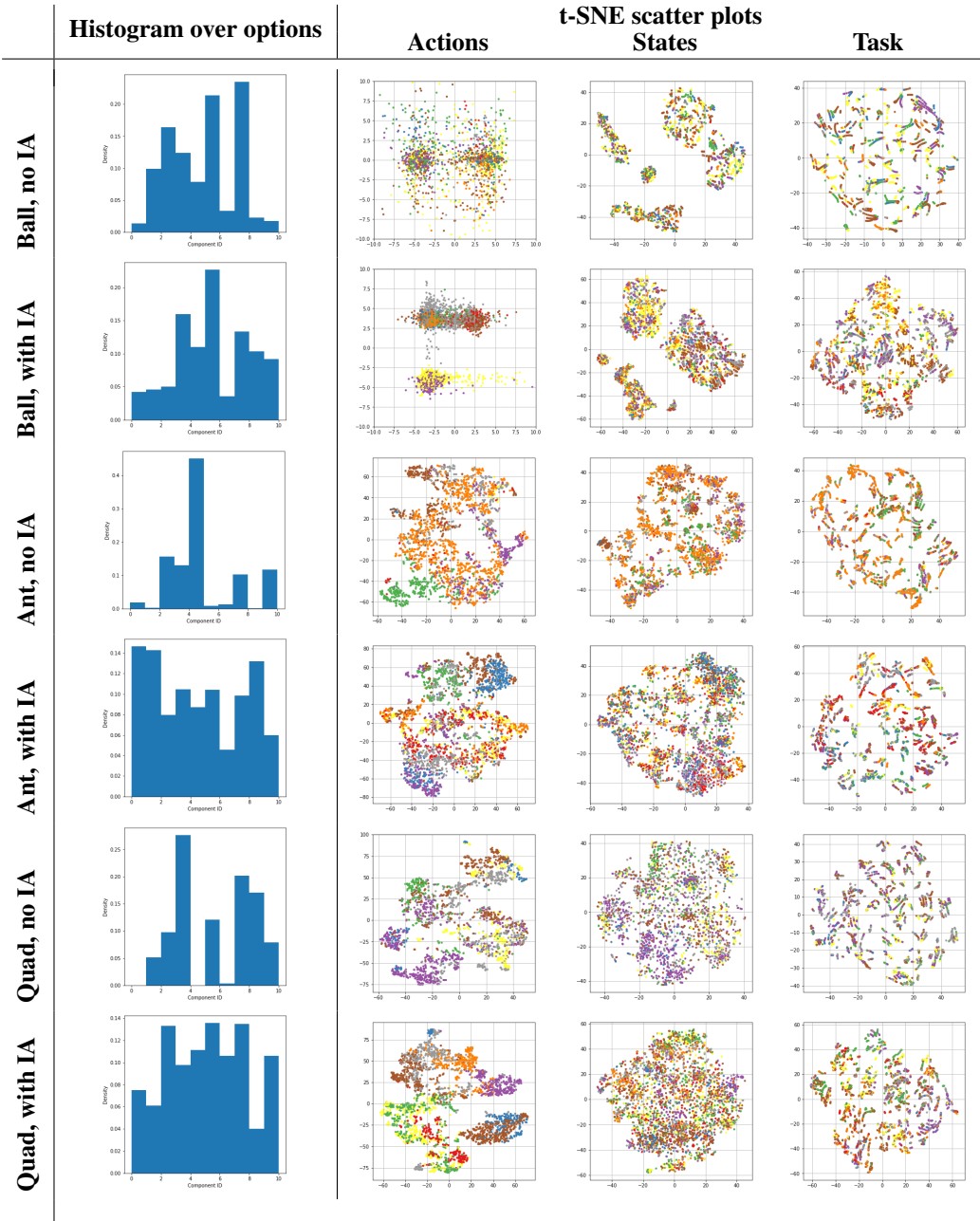

Figure 10: Qualitative results for the three bodies (Ball, Ant, Quad) without limited switches, both with and without IA, obtained over 100 evaluation episodes. **Left**: the histogram over different options used by each agent; **Centre to right**: scatter plots of the action space, state space, and task information, colored by the corresponding option selected. Each of these spaces has been projected to $2D$ using t-SNE, except for the two-dimensional action space for Ball, which is plotted directly. For each case, care has been taken to choose a median / representative model out of 5 seeds.

## A.2 ACTION AND TEMPORAL ABSTRACTION EXPERIMENTS

The complete results for all pixel and proprioception based multitask experiments for ball-in-cup and stacking can be respectively found in Figures 11 and 12. Both RHPO and HO2 outperform a simple Gaussian policy trained via MPO. HO2 additionally improves performance over mixture policies (RHPO) demonstrating that the ability to learn temporal abstraction proves beneficial in these domains. The difference grows as the task complexity increases and is particularly pronounced for the final stacking tasks.

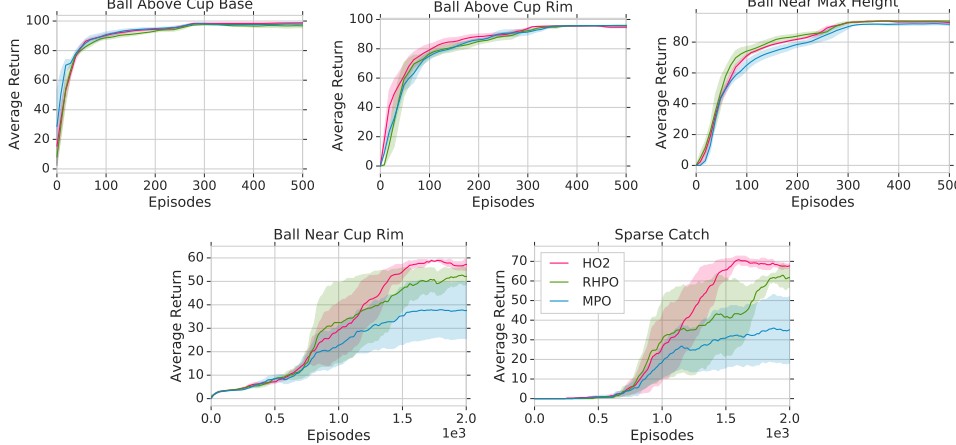

Figure 11: Complete results on pixel-based ball-in-cup experiments.

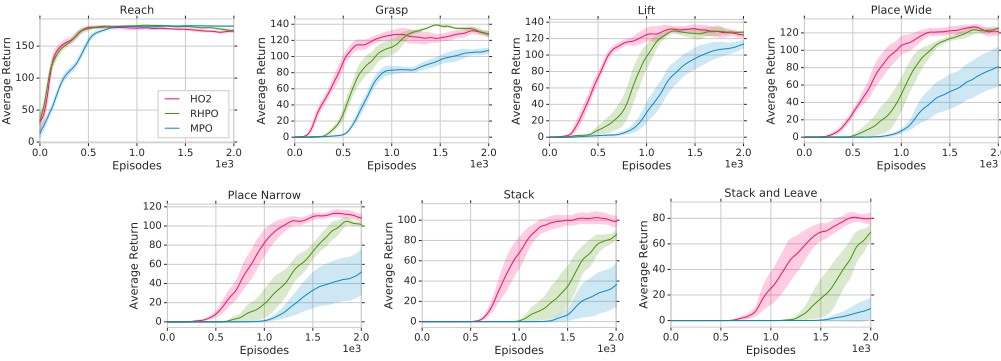

Figure 12: Complete results on pixel-based stacking experiments.

## A.3 TASK-AGNOSTIC TERMINATIONS

The complete results for all experiments with task-agnostic terminations can be found in Figure 13.

The perspective of options as task-independent skills with termination conditions as being part of a skill, leads to termination conditions which are also task independent. We show that at least in this limited set of experiments, the perspective of task-dependent termination conditions - i.e. with access to task information - which can be understood as part of the high-level control mechanism for activating options improves performance. Intuitively, by removing task information from the termination conditions, we constrain the space of solutions which first accelerates training slightly but limits final performance. It additionally shows that while we benefit when sharing options across tasks, each task gains from controlling the length of these options independently. Based on these results, the termination conditions across all other multi-task experiments are conditioned on the active task.

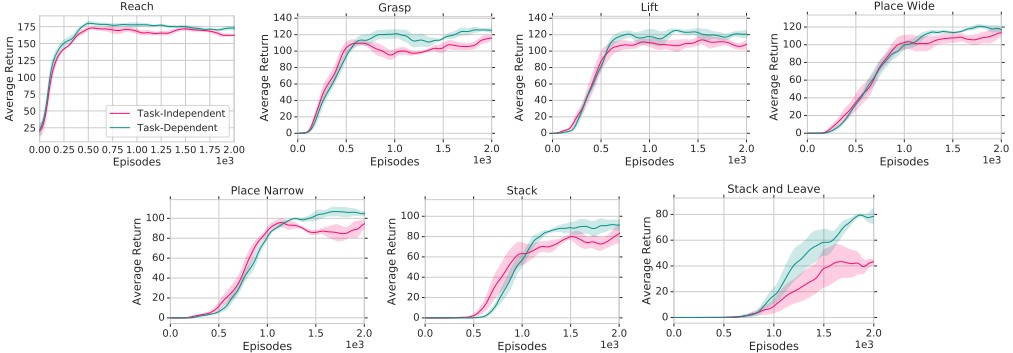

Figure 13: Complete results on multi-task block stacking with and without conditioning termination conditions on tasks.

## A.4 OFF-POLICY OPTION LEARNING

The complete results for all experiments with and without action-conditional inference procedure can be found in Figure 14.

In order to train in a more on-policy regime, we reduce the size of the replay buffer by two orders of magnitude and increase the ratio between data generation (actor steps) and data fitting (learner steps) by one order of magnitude. The resulting algorithm is run without any additional hyperparameter tuning to provide an insight into the effect of conditioning on action probabilities under options in the inference procedure. We can see that in the on-policy case the impact of this change is less pronounced. Across all cases, we were unable to generate significant performance gains by including action conditioning into the inference procedure.

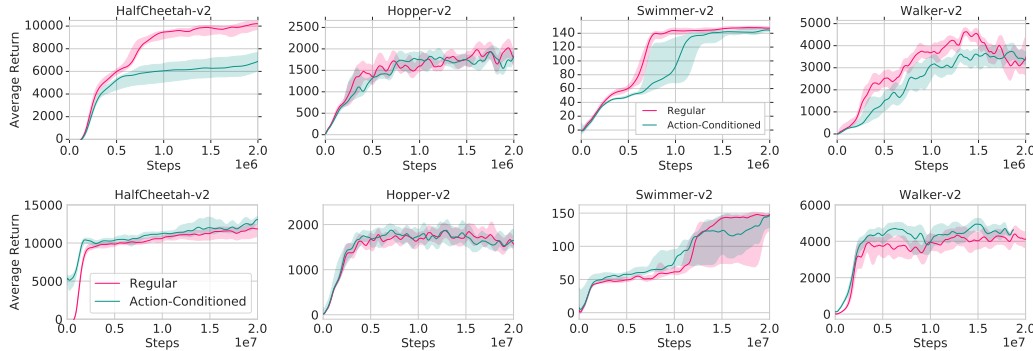

Figure 14: Complete results on OpenAI gym with and without conditioning component probabilities on past executed actions. For the off-policy (top) and on-policy case (bottom). The on-policy approaches uses data considerably less efficiently and the x-axis is correspondingly adapted.

## A.5 TRUST-REGION CONSTRAINTS

The complete results for all trust-region ablation experiments can be found in Figure 15.

With the exception of very high or very low constraints, the approach trains robustly, but performance drops considerably when we remove the constraint fully.

## A.6 SINGLE TIME-STEP VS MULTI TIME-STEP INFERENCE

To investigate the impact of probabilistic inference of posterior option distributions $\pi_H(o_t|h_t)$ along the whole sampled trajectory instead of using sampling-based approximations until the current timestep, we perform additional ablations displayed in Figure 16. Note that we are required to

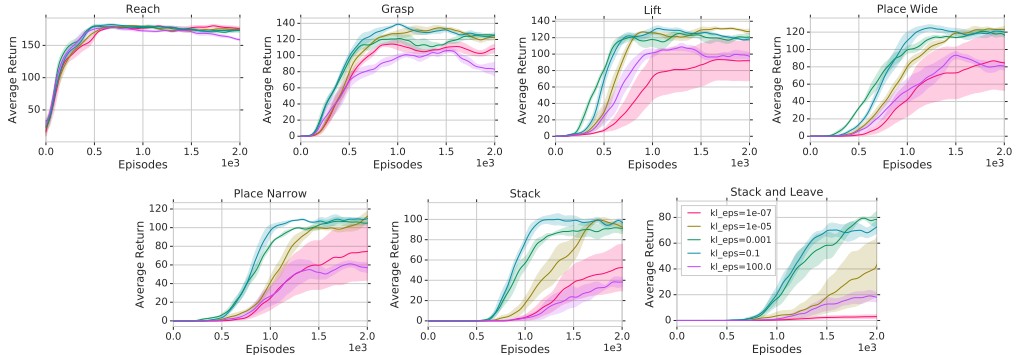

Figure 15: Complete results on block stacking with varying trust-region constraints for both termination conditions $\beta$ and the high-level controller $\pi_C$.

perform probabilistic inference for at least one step to use backpropagation through the inference step to update our policy components. Any completely sampling-based approach would require a different policy optimizer (e.g. via likelihood ratio or raparametrization trick) which would introduce additional compounding effects.

We compare HO2 with an ablation of HO2 where we do not compute the option probabilities along the trajectory following Equation 4 but instead use an approximation with only concrete option samples propagating across timesteps for all steps until the current step. For generating action samples, we therefore sample options for every timestep along a trajectory without keeping a complete distribution over options and sample actions only from the active option at every timestep. For determining the likelihood of actions and options for every timestep, we rely on Equation 2 based the sampled options of the previous timestep. By using samples and the critic-weighted update procedure from Equation 8, we can only generate gradients for the policy for the current timestep instead of backpropagating through the whole inference procedure. We find that using both samples from executed options reloaded from the buffer as well as new samples during learning both can reduce performance depending on the domain. However, in the Hopper-v2 environment, sampling during learning performs slightly better than inferring options.

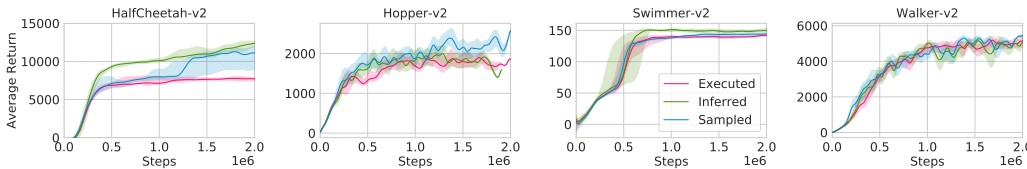

Figure 16: Ablation results comparing inferred options with sampled options during learning (sampled) and during execution (executed). The ablation is run with five actors instead of a single one as used in the OpenAI gym experiments in order to generate results faster.

## B  ADDITIONAL EXPERIMENT DETAILS

### B.1  OPENAI GYM EXPERIMENTS

All experiments are run with asynchronous learner and actors. We use a single actor and report performance over the number of transitions generated. Following (Wulfmeier et al., 2020), both HO2 and RHPO use different biases for the initial mean of all options or mixture components - distributed between minimum and maximum action output. This provides a small but non-negligible benefit and supports specialization of individual options. In line with our baselines (DAC (Zhang & Whiteson, 2019), IOPG (Smith et al., 2018), Option Critic (Bacon et al., 2017)) we use 4 options or mixture components for the OpenAI gym experiments. We run all experiments with 5 samples and report variance and mean. All experiments are run with a single actor in a distributed setting.

| Hyperparameters | HO2 | RHPO | MPO |
|---|---|---|---|
| Policy net | | 256-256 | |
| Number of actions sampled per state | | 20 | |
| Q function net | | 256-256 | |
| Number of components | | 4 | NA |
| $\epsilon$ | | 0.1 | |
| $\epsilon_\mu$ | | 5e-4 | |
| $\epsilon_\Sigma$ | | 5e-5 | |
| $\epsilon_\alpha$ | | 1e-4 | NA |
| $\epsilon_t$ | 1e-4 | NA | |
| Discount factor ($\gamma$) | | 0.99 | |
| Adam learning rate | | 3e-4 | |
| Replay buffer size | | 2e6 | |
| Target network update period | | 200 | |
| Batch size | | 256 | |
| Activation function | | elu | |
| Layer norm on first layer | | Yes | |
| Tanh on output of layer norm | | Yes | |
| Tanh on input actions to Q-function | | Yes | |
| Sequence length | | 8 | |

Table 2: Hyperparameters - OpenAI gym

## B.2 ACTION AND TEMPORAL ABSTRACTION EXPERIMENTS

Shared across all algorithms, we use 3-layer convolutional policy and Q-function torsos with [128, 64, 64] feature channels, [(4, 4), (3, 3), (3, 3)] as kernels and stride 2. For all multitask domains, we build on information asymmetry and only provide task information as input to the high-level controller and termination conditions to create additional incentive for the options to specialize. The Q-function has access to all observations (see the corresponding tables in this section). We follow (Riedmiller et al., 2018; Wulfmeier et al., 2020) and assign rewards for all possible tasks to trajectories when adding data to the replay buffer independent of the generating policy.

**Stacking** The setup consists of a Sawyer robot arm mounted on a table and equipped with a Robotiq 2F-85 parallel gripper. In front of the robot there is a basket of size 20x20 cm which contains three cubes with an edge length of 5 cm (see Figure 4).

The agent is provided with proprioception information for the arm (joint positions, velocities and torques), and the tool center point position computed via forward kinematics. For the gripper, it receives the motor position and velocity, as well as a binary grasp flag. It also receives a wrist sensor's force and torque readings. Finally, it is provided with three RGB camera images at $64 \times 64$ resolution. At each timestep, a history of two previous observations (except for the images) is provided to the agent, along with the last two joint control commands. The observation space is detailed in Table 5. All stacking experiments are run with 50 actors in parallel and reported over the current episodes generated by any actor. Episode lengths are up to 600 steps.

The robot arm is controlled in Cartesian velocity mode at 20Hz. The action space for the agent is 5-dimensional, as detailed in Table 4. The gripper movement is also restricted to a cubic volume above the basket using virtual walls.

$$stol(v, \epsilon, r) = \begin{cases} 1 & \text{iff } |v| < \epsilon \\ 1 - tanh^2(\frac{atanh(\sqrt{0.95})}{r}|v|) & \text{else} \end{cases} \quad (10)$$

$$slin(v, \epsilon_{min}, \epsilon_{max}) = \begin{cases} 0 & \text{iff } v < \epsilon_{min} \\ 1 & \text{iff } v > \epsilon_{max} \\ \frac{v - \epsilon_{min}}{\epsilon_{max} - \epsilon_{min}} & \text{else} \end{cases} \quad (11)$$

| Hyperparameters | HO2 | RHPO | MPO |
|---|---|---|---|
| Policy torso (shared across tasks) | 256 | | 512 |
| Policy task-dependent heads | 100 (categorical) | | 200 |
| Policy shared heads | 100 (components) | | NA |
| Policy task-dependent terminations | 100 (terminations) | NA | NA |
| $\epsilon_\mu$ | 1e-3 | | |
| $\epsilon_\Sigma$ | 1e-5 | | |
| $\epsilon_\alpha$ | 1e-4 | | NA |
| $\epsilon_t$ | 1e-4 | NA | |
| Number of action samples | 20 | | |
| Q function torso (shared across tasks) | 400 | | |
| Q function head (per task) | 300 | | |
| Number of components | number of tasks | | NA |
| Replay buffer size | 1e6 | | |
| Target network update period | 500 | | |
| Batch size | 256 | | |

Table 3: Hyperparameters. Values are taken from the OpenAI gym experiments with the above mentioned changes.

Table 4: Action space for the Sawyer Stacking experiments.

| Entry | Dims | Unit | Range |
|---|---|---|---|
| Translational Velocity in x, y, z | 3 | m/s | [-0.07, 0.07] |
| Wrist Rotation Velocity | 1 | rad/s | [-1, 1] |
| Finger speed | 1 | tics/s | [-255, 255] |

Table 5: Observations for the Sawyer Stacking experiments. The TCP's pose is represented as its world coordinate position and quaternion. In the table, $m$ denotes meters, $rad$ denotes radians, and $q$ refers to a quaternion in arbitrary units ($au$).

| Entry | Dims | Unit | History |
|---|---|---|---|
| Joint Position (Arm) | 7 | rad | 2 |
| Joint Velocity (Arm) | 7 | rad/s | 2 |
| Joint Torque (Arm) | 7 | Nm | 2 |
| Joint Position (Hand) | 1 | tics | 2 |
| Joint Velocity (Hand) | 1 | tics/s | 2 |
| Force-Torque (Wrist) | 6 | N, Nm | 2 |
| Binary Grasp Sensor | 1 | au | 2 |
| TCP Pose | 7 | m, au | 2 |
| Camera images | $3 \times 64 \times 64 \times 3$ | R/G/B value | 0 |
| Last Control Command | 8 | rad/s, tics/s | 2 |

$$btol(v, \epsilon) = \begin{cases} 1 & \text{iff } |v| < \epsilon \\ 0 & \text{else} \end{cases} \qquad (12)$$

- *REACH(G)*: $stol(d(TCP, G), 0.02, 0.15)$:
  Minimize the distance of the TCP to the green cube.

- *GRASP*:
  Activate grasp sensor of gripper ("inward grasp signal" of Robotiq gripper)
- *LIFT(G)*: $slin(G, 0.03, 0.10)$
  Increase z coordinate of an object more than 3cm relative to the table.
- *PLACE_WIDE(G, Y)*: $stol(d(G, Y + [0, 0, 0.05]), 0.01, 0.20)$
  Bring green cube to a position 5cm above the yellow cube.
- *PLACE_NARROW(G, Y)*: $stol(d(G, Y + [0, 0, 0.05]), 0.00, 0.01)$:
  Like PLACE_WIDE(G, Y) but more precise.
- *STACK(G, Y)*: $btol(d_{xy}(G, Y), 0.03) * btol(d_z(G, Y) + 0.05, 0.01) * (1 - GRASP)$
  Sparse binary reward for bringing the green cube on top of the yellow one (with 3cm tolerance horizontally and 1cm vertically) and disengaging the grasp sensor.
- *STACK_AND_LEAVE(G, Y)*: $stol(d_z(TCP, G) + 0.10, 0.03, 0.10) * STACK(G, Y)$
  Like STACK(G, Y), but needs to move the arm 10cm above the green cube.

**Ball-In-Cup** This task consists of a Sawyer robot arm mounted on a pedestal. A partially see-through cup structure with a radius of 11cm and height of 17cm is attached to the wrist flange. Between cup and wrist there is a ball bearing, to which a yellow ball of 4.9cm diameter is attached via a string of 46.5cm length (see Figure 4).

Most of the settings for the experiment align with the stacking task. The agent is provided with proprioception information for the arm (joint positions, velocities and torques), and the tool center point and cup positions computed via forward kinematics. It is also provided with two RGB camera images at $64 \times 64$ resolution. At each timestep, a history of two previous observations (except for the images) is provided to the agent, along with the last two joint control commands. The observation space is detailed in Table 7. All BIC experiments are run with 20 actors in parallel and reported over the current episodes generated by any actor. Episode lengths are up to 600 steps.

The position of the ball in the cup's coordinate frame is available for reward computation, but not exposed to the agent. The robot arm is controlled in joint velocity mode at 20Hz. The action space for the agent is 4-dimensional, with only 4 out of 7 joints being actuated, in order to avoid self-collision. Details are provided in Table 4.

Table 6: Action space for the Sawyer Ball-in-Cup experiments.

| Entry | Dims | Unit | Range |
|---|---|---|---|
| Rotational Joint Velocity for joints 1, 2, 6 and 7 | 4 | rad/s | [-2, 2] |

Table 7: Observations for the Sawyer Ball-in-Cup experiments. In the table, $m$ denotes meters, $rad$ denotes radians, and $q$ refers to a quaternion in arbitrary units ($au$). Note: the joint velocity and command represent the robot's internal state; the 3 degrees of freedom that were fixed provide a constant input of 0.

| Entry | Dims | Unit |
|---|---|---|
| Joint Position (Arm) | 7 | rad |
| Joint Velocity (Arm) | 7 | rad/s |
| TCP Pose | 7 | m, au |
| Camera images | $2 \times 64 \times 64 \times 3$ | R/G/B value |
| Last Control Command | 7 | rad/s |

Let $B_A$ be the Cartesian position in meters of the ball in the cup's coordinate frame (with an origin at the center of the cup's bottom), along axes $A \in \{x, y, z\}$.

- *CATCH*: $0.17 > B_z > 0$ and $||B_{xy}||_2 < 0.11$
  Binary reward if the ball is inside the volume of the cup.
- *BALL_ABOVE_BASE*: $B_z > 0$
  Binary reward if the ball is above the bottom plane of the cup.

- *BALL_ABOVE_RIM*: $B_z > 0.17$
  Binary reward if the ball is above the top plane of the cup.
- *BALL_NEAR_MAX*: $B_z > 0.3$
  Binary reward if the ball is near the maximum possible height above the cup.
- *BALL_NEAR_RIM*: $1 - tanh^2(\frac{atanh(\sqrt{0.95})}{0.5} \times ||B_{xyz} - (0,0,0.17)||_2)$
  Shaped distance of the ball to the center of the cup opening (0.95 loss at a distance of 0.5).

### B.3 PRE-TRAINING AND SEQUENTIAL TRANSFER EXPERIMENTS

The sequential transfer experiments are performed with the same settings as their multitask equivalents. However, they rely on a pre-training step in which we take all but the final task in each domain and train HO2 to pre-train options which we then transfer with new high-level controller on the final task. Fine-tuning of the options is enabled as we find that it produces slightly better performance. Only data used for the final training step is reported but all both approaches were trained for the same amount of data during pretraining until convergence.

### B.4 LOCOMOTION EXPERIMENTS

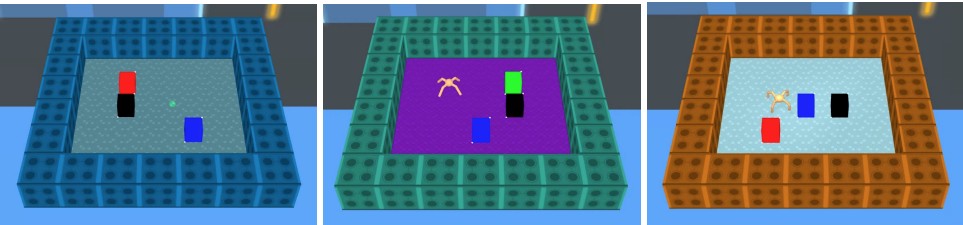

Figure 17: The environment used for simple locomotion tasks with Ball (left), Ant (center) and Quadruped (right).

Figure 17 shows examples of the environment for the different bodies used. In addition to proprioceptive agent state information (which includes the body height, position of the end-effectors, the positions and velocities of its joints and sensor readings from an accelerometer, gyroscope and velocimeter attached to its torso), the state space also includes the ego-centric coordinates of all target locations and a categorical index specifying the task of interest. Table 8 contains an overview of the observations and action dimensions for this task. The agent receives a sparse reward of $+60$ if part of its body reaches a square surrounding the predicate location, and 0 otherwise. Both the agent spawn location and target locations are randomized at the start of each episode, ensuring that the agent must use both the task index and target locations to solve the task.

Table 8: Observations for the *go to one of 3 targets* task with 'Ball' and 'Ant.

| Entry | Dimensionality |
|---|---|
| Task Index | 3 |
| Target locations | 9 |
| Proprioception (Ball) | 16 |
| Proprioception (Ant) | 41 |
| Proprioception (Quad) | 57 |
| Action Dim (Ball) | 2 |
| Action Dim (Ant) | 8 |
| Action Dim (Quad) | 12 |

## C ADDITIONAL DERIVATIONS

In this section we explain the derivations for training option policies with options parameterized as Gaussian distributions. Each policy improvement step is split into two parts: non-parametric and parametric update.

## C.1   NON-PARAMETRIC OPTION POLICY UPDATE

In order to obtain the non-parametric policy improvement we optimize the following equation:

$$\max_q \mathbb{E}_{h_t \sim p(h_t)}\big[\mathbb{E}_{a_t,o_t \sim q}\big[Q_\phi(s_t, a_t, o_t)\big]\big]$$

$$s.t. \mathbb{E}_{h_t \sim p(h_t)}\big[\mathrm{KL}(q(\cdot|h_t), \pi_\theta(\cdot|h_t))\big] < \epsilon_E$$

$$s.t. \mathbb{E}_{h_t \sim p(h_t)}\big[\mathbb{E}_{q(a_t,o_t|h_t)}\big[1\big]\big] = 1.$$

for each step $t$ of a trajectory, where $h_t = \{s_t, a_{t-1}, s_{t-1}, ...a_0, s_0\}$ represents the history of states and actions and $p(h_t)$ describes the distribution over histories for timestep $t$, which in practice are approximated via the use of a replay buffer $\mathcal{D}$. When sampling $h_t$, the state $s_t$ is the first element of the history. The inequality constraint describes the maximum allowed KL distance between intermediate update and previous parametric policy, while the equality constraint simply ensures that the intermediate update represents a normalized distribution.

Subsequently, in order to render the following derivations more intuitive, we replace the expectations and explicitly use integrals. The Lagrangian $L(q, \eta, \gamma)$ can now be formulated as

$$L(q,\eta,\gamma) = \iiint p(h_t)q(a_t, o_t|h_t)Q_\phi(s_t, a_t, o_t)\, \mathrm{d}o_t\, \mathrm{d}a_t\, \mathrm{d}h_t \tag{13}$$

$$+ \eta\left(\epsilon_E - \iiint p(h_t)q(a_t, o_t|h_t)\log\frac{q(a_t, o_t|h_t)}{\pi_\theta(a_t, o_t|h_t)}\, \mathrm{d}o_t\, \mathrm{d}a_t\, \mathrm{d}h_t\right) \tag{14}$$

$$+ \gamma\left(1 - \iiint p(h_t)q(a_t, o_t|h_t)\, \mathrm{d}o_t\, \mathrm{d}a_t\, \mathrm{d}h_t\right). \tag{15}$$

Next to maximize the Lagrangian with respect to the primal variable $q$, we determine its derivative as,

$$\frac{\partial L(q,\eta,\gamma)}{\partial q} = Q_\phi(a_t, o_t, s_t) - \eta\log q(a_t, o_t|h_t) + \eta\log \pi_\theta(a_t, o_t|h_t) - \eta - \gamma.$$

In the next step, we can set the left hand side to zero and rearrange terms to obtain

$$q(a_t, o_t|h_t) = \pi_\theta(a_t, o_t|h_t)\exp\left(\frac{Q_\phi(s_t, a_t, o_t)}{\eta}\right)\exp\left(-\frac{\eta + \gamma}{\eta}\right).$$

The last exponential term represents a normalization constant for $q$, which we can formulate as

$$\frac{\eta + \gamma}{\eta} = \log\left(\iint \pi_\theta(a_t, o_t|h_t)\exp\left(\frac{Q_\phi(s_t, a_t, o_t)}{\eta}\right)\, \mathrm{d}o_t\, \mathrm{d}a_t\right). \tag{16}$$

In order to obtain the dual function $g(\eta)$, we insert the solution for the primal variable into the Lagrangian in Equation 13 which yields

$$L(q,\eta,\gamma) = \iiint p(h_t)q(a_t, o_t|h_t)Q_\phi(s_t, a_t, o_t)\, \mathrm{d}o_t\, \mathrm{d}a_t\, \mathrm{d}h_t$$

$$+ \eta\left(\epsilon_E - \iiint p(h_t)q(a_t, o_t|h_t)\log\frac{\pi_\theta(a_t, o_t|h_t)\exp\left(\frac{Q_\phi(s_t, a_t, o_t)}{\eta}\right)\exp\left(-\frac{\eta+\gamma}{\eta}\right)}{\pi_\theta(a_t, o_t|h_t)}\, \mathrm{d}o_t\, \mathrm{d}a_t\, \mathrm{d}h_t\right)$$

$$+ \gamma\left(1 - \iiint p(h_t)q(a_t, o_t|h_t)\, \mathrm{d}o_t\, \mathrm{d}a_t\, \mathrm{d}h_t\right).$$

We expand the equation and rearrange to obtain

$$
\begin{aligned}
L(q, \eta, \gamma) =& \iiint p(h_t) q(a_t, o_t | h_t) Q_\phi(s_t, a_t, o_t) \, \mathrm{d}o_t \, \mathrm{d}a_t \, \mathrm{d}h_t \\
& - \eta \iiint p(h_t) q(a_t, o_t | h_t) \left[ \frac{Q_\phi(s_t, a_t, o_t)}{\eta} + \log \pi_\theta(a_t, o_t | h_t) - \frac{\eta + \gamma}{\eta} \right] \mathrm{d}o_t \, \mathrm{d}a_t \, \mathrm{d}h_t \\
& + \eta \epsilon_E + \eta \iiint p(h_t) q(a_t, o_t | h_t) \log \pi_\theta(a_t, o_t | h_t) \, \mathrm{d}o_t \, \mathrm{d}a_t \, \mathrm{d}h_t \\
& + \gamma \left( 1 - \iiint p(h_t) q(a_t, o_t | h_t) \, \mathrm{d}o_t \, \mathrm{d}a_t \, \mathrm{d}h_t \right).
\end{aligned}
$$

In the next step, most of the terms cancel out and after additional rearranging of the terms we obtain

$$
L(q, \eta, \gamma) = \eta \epsilon_E + \eta \int p(h_t) \frac{\eta + \gamma}{\eta} \, \mathrm{d}h_t.
$$

We have already calculated the term inside the integral in Equation 16, which we now insert to obtain

$$
\begin{aligned}
g(\eta) &= \min_q L(q, \eta, \gamma) \\
&= \eta \epsilon_E + \eta \int p(h_t) \log \left( \iint \pi_\theta(a_t, o_t | h_t) \exp \left( \frac{Q_\phi(s_t, a_t, o_t)}{\eta} \right) \mathrm{d}o_t \, \mathrm{d}a_t \right) \mathrm{d}h_t \\
&= \eta \epsilon_E + \eta \mathbb{E}_{h_t \sim p(h_t)} \left[ \log \left( \mathbb{E}_{a_t, o_t \sim \pi_\theta} \left[ \exp \left( \frac{Q_\phi(s_t, a_t, o_t)}{\eta} \right) \right] \right) \right],
\end{aligned}
\tag{17}
$$

The dual in Equation 17 can finally be minimized with respect to $\eta$ based on samples from the replay buffer and policy.

## C.2 PARAMETRIC OPTION POLICY UPDATE

After obtaining the non-parametric policy improvement, we can align the parametric option policy to the current non-parametric policy. As the non-parametric policy is represented by a set of samples from the parametric policy with additional weighting, this step effectively employs a type of critic-weighted maximum likelihood estimation. In addition, we introduce regularization based on a distance function $\mathcal{T}$ which has a trust-region effect for the update and stabilizes learning.

$$
\begin{aligned}
\theta_{new} &= \arg \min_\theta \mathbb{E}_{h_t \sim p(h_t)} \left[ \mathrm{KL}\big( q(a_t, o_t | h_t) \| \pi_\theta(a_t, o_t | h_t) \big) \right] \\
&= \arg \min_\theta \mathbb{E}_{h_t \sim p(h_t)} \left[ \mathbb{E}_{a_t, o_t \sim q} \left[ \log q(a_t, o_t | h_t) - \log \pi_\theta(a_t, o_t | h_t) \right] \right] \\
&= \arg \max_\theta \mathbb{E}_{h_t \sim p(h_t), a_t, o_t \sim q} \left[ \log \pi_\theta(a_t, o_t | h_t) \right], \\
&\text{s.t. } \mathbb{E}_{h_t \sim p(h_t)} \left[ \mathcal{T}(\pi_{\theta_{new}}(\cdot | h_t) | \pi_\theta(\cdot | h_t)) \right] < \epsilon_M,
\end{aligned}
$$

where $h_t \sim p(h_t)$ is a trajectory segment, which in practice sampled from the dataset $\mathcal{D}$, $\mathcal{T}$ is an arbitrary distance function between the new policy and the previous policy. $\epsilon_M$ denotes the allowed change for the policy. We again employ Lagrangian Relaxation to enable gradient based optimization of the objective, yielding the following primal:

$$
\begin{aligned}
\max_\theta \min_{\alpha > 0} L(\theta, \alpha) =& \mathbb{E}_{h_t \sim p(h_t), a_t, o_t \sim q} \left[ \log \pi_\theta(a_t, o_t | h_t) \right] + \\
& \alpha \left( \epsilon_M - \mathbb{E}_{h_t \sim p(h_t)} \left[ \mathcal{T}(\pi_{\theta_{new}}(\cdot | h_t), \pi_\theta(\cdot | h_t)) \right] \right).
\end{aligned}
\tag{18}
$$

We can solve for $\theta$ by iterating the inner and outer optimization programs independently. In practice we find that it is most efficient to update both in parallel.

We also define the following distance function between old and new option policies

$$\mathcal{T}(\pi_{\theta_{new}}(\cdot|h_t), \pi_\theta(\cdot|h_t)) = \mathcal{T}_H(h_t) + \mathcal{T}_T(h_t) + \mathcal{T}_L(h_t)$$

$$\mathcal{T}_H(h_t) = \text{KL}(\text{Cat}(\{\alpha^j_{\theta_{new}}(h_t)\}_{j=1...M}) \| \text{Cat}(\{\alpha^j_\theta(h_t)\}_{j=1...M}))$$

$$\mathcal{T}_T(h_t) = \frac{1}{M}\sum_{j=1}^M \text{KL}(\text{Cat}(\{\beta^{ij}_{\theta_{new}}(h_t)\}_{j=1...2}) \| \text{Cat}(\{\beta^{ij}_\theta(h_t)\}_{j=1...2}))$$

$$\mathcal{T}_L(h_t) = \frac{1}{M}\sum_{j=1}^M \text{KL}(\mathcal{N}(\mu^j_{\theta_{new}}(h_t), \Sigma^j_{\theta_{new}}(h_t)) \| \mathcal{N}(\mu^j_\theta(h_t), \Sigma^j_\theta(h_t)))$$

where $\mathcal{T}_H$ evaluates the KL between the categorical distributions of the high-level controller, $\mathcal{T}_T$ is the average KL between the categorical distributions of the all termination conditions, and $\mathcal{T}_L$ corresponds to the average KL across Gaussian components. In practice, we can exert additional control over the convergence of model components by applying different $\epsilon_M$ to different model parts (high-level controller, termination conditions, options).

## C.3 TRANSITION PROBABILITIES FOR OPTION AND SWITCH INDICES

The transitions for option $o$ and switch index $n$ are given by:

$$p(o_t, n_t | s_t, o_{t-1}, n_{t-1}) = \begin{cases} (1 - \beta(s_t, o_{t-1})) & \text{if } n_t = n_{t-1}, o_t = o_{t-1} \\ \beta(s_t, o_{t-1})\pi^C(o_t|s_t) & \text{if } n_t = n_{t-1} + 1 \\ 0 & \text{otherwise} \end{cases} \tag{19}$$

