# OpenReview forum: "Data-efficient Hindsight Off-policy Option Learning"
_ICLR.cc/2021/Conference — Reject_

### Official Review · AnonReviewer1 · 2020-10-20
**Very interesting paper but difficult to follow**

**Rating:** 5
**Confidence:** 4

**Review:**

The paper considers the Hierarchical Reinforcement Learning setting, Options in particular, and proposes an algorithm that allows to learn both the high-level and low-level (option) policies at once, from off-policy samples. An original aspect of the algorithm is that it is easy to constrain the learned policies on how often they terminate an option and start a new one. This prevents the agent from learning tiny options that immediately terminate. It is unclear whether it can also be used to prevent the agent from learning a single big option that does everything.

The paper is very interesting and has a high educational value. It combines many different approaches and is mathematically sound. The empirical evaluation shows encouraging results. However, the significance of the empirical results is difficult to measure, due to a few cons of this paper:

- The paper is quite difficult to follow, and requires several attentive reads to be understood (by someone having a deep knowledge about options, intra-option learning and the Option-Critic architecture). I believe that the lack of clarity comes from the brevity of the paper, that has to fit in the page limit. I would suggest the authors to remove Figures 1 and 2, that take place while still being very difficult to understand. The text helps understand the figures, the figures do not help understand the text, so I would remove the figures. "Problem setup" in the preliminaries could also be removed, and replaced with an introductory paragraph in "Method", that clearly states what the contribution will be, and what are its main components/properties.
- "multi-task learning", just before "Experiments", is then used in the experiments to increase the sample-efficiency of all the algorithms. Because it is used for all the algorithms and not ablated, I don't see how it contributes to the paper. I think that the paper already proposes many ideas, and that multi-task learning could be omitted and left for a future paper.
- In the experiments, comparing MPO with HO2 allows to see a benefit from the use of options, with the proposed learning algorithm. This is a good point. However, MPO is not a well-known baseline, and a quick glance at the plots does not allow the reader to see that MPO does not use options. I would suggest either to add a little "no options" next to MPO in the figures, or to replace it (or add to it) PPO, ACKTR, ACER or the Soft Actor-Critic (SAC). These algorithms are well-known baselines. Not all of them are off-policy, but I believe that comparing HO2 to state-of-the-art algorithms, without restriction to off-policy ones, would better allow to illustrate all the gains to be obtained by HO2.
- A good argument for the use of options is to use them for explainability: telling an observer what is the current option, and what it tries to achieve. Is it possible to have a small discussion of what do the options learned by HO2 do? Do they aim at goals, or do they perform small bits of trajectories?

In summary, I like the proposed algorithm, the core contribution of the paper, the contents of Section 3. However, the clarity of this paper is to me too low, and prevents fully grasping the impact of the proposed research. I therefore recommend against acceptance, and invite the authors to remove parts on multi-task, figures that do not help, and use well-known baselines in their evaluation. With the space gained with these changes, more text can be spent summarizing what the contribution will be, and motivating the use of options.

Author response: the authors answered my question about the absence of learning curves, and provided extra details. However, I still think that the paper could be clearer and more focused, a sentiment that I think I share with the other reviewers. Given my hesitation, I would therefore not vote for accepting this paper, but I acknowledge that the proposed method is original and interesting, so I would not mind if this paper were to be accepted.

---

> ### Author Response · Authors · 2020-11-17
> **Individual feedback per review**
>
> We thank the reviewer for their detailed and constructive feedback. In this post, we will focus on reviewer specific feedback. We will additionally provide an overview post describing the general feedback and corresponding changes.
>
> - Motivating the use of options
>
> We thank the reviewer for raising this point. With the additional page available to address such concerns, we have extended the introduction to more clearly motivate options and discuss the merits of such an approach before delving into the existing options literature.
>
> - Lack of clarity (figures, contributions, and multi-task learning)
>
> Figures 1 and 2 are used to support understanding of the option policies, mixture policies and temporal consistency parts in Section 3 by providing the corresponding graphical models. We believe that this will help understanding for readers who are more familiar with graphical models, sequence modelling, and related fields, while those more familiar with the option literature may benefit more from the derivations in Section 3. We have added additional references in the text when introducing the corresponding equations, improved the captions, and added some brief intuition to the start of each subsection.
>
> Our contributions are explicitly stated in bullet form at the end of Section 1. We additionally separate the method section with respect to flat Gaussians and mixture policies which have been trained via critic-weighted maximum likelihood in prior work and the proposed extension to train option policies. The current writing aims to find a compromise between clear separation of methods and clarity and self-containment of the method description. We have furthermore emphasized the use of MPO and RHPO to train Gaussian and mixture policies in prior work.
>
> Finally, since options represent reusable behaviour abstractions, they are known to be beneficial in a multi-task setting where they can be shared across tasks (see e.g. [1]). We additionally use multitask learning with related tasks to enable us to address more complex domains with acceptable data requirements. The described methods for multi-task learning have been proposed and tested in prior work and we merely use them here to accelerate learning. We have integrated the corresponding description into the experimental section and adapted it for clarification.
>
> - Clarity regarding the use of MPO
>
> We use MPO (using flat Gaussian policies) and RHPO (using mixture policies) as the baseline methods for comparison with HO2, because they use an equivalent underlying optimization procedure based on critic-reweighted likelihood estimation. The fact that the base algorithm is equivalent makes it easy to compare the three approaches and to isolate the effects of action abstraction (which MPO does not have) and temporal abstraction (which both MPO and RHPO miss). We have extended the text to make this clear, and have also clarified that MPO trains a flat policy in the experiments, as suggested by the reviewer.
>
> - On learning a single degenerate option that solves everything
>
> The reviewer is right to point out that in certain cases, hierarchical approaches can discover degenerate solutions such as a single option. In the case of HO2, the experiments found that with enough structure in the problem setup, a diverse set of options was learned and used - this is also supported and explained by the “Further analysis” experiments in Section 4.3.
>
>
> [Feedback continues in the next post]

---

> > ### Author Response · Authors · 2020-11-17
> > **Continuation**
> >
> > - What do the options do? Do they correspond to goal-directed behaviour or trajectory snippets?
> >
> > This is already explained in the paper in Section 4.3 (in brief) and in Appendix A (in much more detail). We have extended the analysis in the main paper for clarity.
> >
> > To summarise, we applied HO2 to smaller and more interpretable domains, with walkers in go-to-target navigation tasks. We ran experiments with different bodies, different forms of information-asymmetry, and with/without the switching constraint between options.
> > In short, we find that the options can represent both task-directed behaviours or short trajectories of motor behaviours (such as moving in certain directions). Additionally, information-asymmetry between high- and low-level controllers can lead to greater diversity of options as we demonstrate in Section 4.3.
> > On the related point of switch constraints: as the reviewer correctly identifies, they can be used to encourage temporal consistency. Even without these constraints, the options are already relatively consistent, with fairly low switch rates (Figure 6). Constraining the frequency of switching increases temporal consistency without hampering the agent’s ability to solve the task and improves performance for transfer with pre-trained options.
> >
> > If not addressed explicitly in the author feedback, we will address the remaining minor comments directly in the paper. Please do not hesitate to emphasise remaining questions or concerns.
> >
> >
> > [1] Bacon, P. L., Harb, J., & Precup, D. (2017, February). The option-critic architecture. In Thirty-First AAAI Conference on Artificial Intelligence.

---

> > > ### Comment · AnonReviewer1 · 2020-11-23
> > > **Better empirical comparison**
> > >
> > > I thank the authors for their detailed response, and welcome their revision of the paper. The response addresses most of my questions, and clarifies a few aspects of the paper. The revised paper now seems clearer, and I like the discussion on option clustering and Figure 9.
> > >
> > > My only (relatively minor) question that remains is why Figure 3 shows horizontal lines for the on-policy baselines (DAC, Option-Critic, IOPG)? These baselines need samples to learn a policy, and I was expecting to see a learning curve (with "steps" being the number of time-steps executed in the environment, in that case). I'm also wondering for how long the baselines have been trained on the tasks, and I did not find that information in the paper (I may have missed it). Are these lines obtained by looking at the results presented in the respective papers of the baselines? I think that stating and motivating why the baselines are well-trained and strong is important to convince the readers that the proposed method outperforms a large variety of related approaches.

---

> > > > ### Author Response · Authors · 2020-11-24
> > > > **Feedback**
> > > >
> > > > We are pleased that the responses were helpful and are certain that addressing the reviewer’s comments has improved the paper considerably.
> > > >
> > > > We have already improved the description of the baselines in the previous update but recognise the need to further clarify. The lines describe the performance after 2*10^6 steps as described in the caption and are taken from [1]. We have confirmed with the authors of the paper that the settings match exactly between the experiments.  We chose to follow this direction to ensure that we take the best known results in this benchmark (a common practice in many fields e.g. computer vision, NLP and partially used in RL as well), instead of using a sub-optimal reimplementation of the algorithms which would potentially underperform existing results.
> > > >
> > > > Using straight lines to indicate final results after the maximum training time of 2x10^6 steps instead of complete learning curves has two reasons. First, to prevent additional clutter in the graphs. Second, the learning curve comparison between on-policy and off-policy learning is only meaningful within limits. While we can align the number of actor steps, we cannot do so for learner steps as the ratio can be independently chosen in off-policy learning.
> > > >
> > > > We have further emphasised these aspects in an updated version of the paper to ensure the readers are aware of this setting.
> > > >
> > > >
> > > > [1] Zhang, S., & Whiteson, S. (2019). DAC: The double actor-critic architecture for learning options. In Advances in Neural Information Processing Systems (pp. 2012-2022).

---

### Official Review · AnonReviewer3 · 2020-10-25
**This paper proposes an efficient option learning method based on TD(0) type objective. The overall objective relies on both action abstraction and temporal abstraction, an interesting ablation study is given to understand more the effect of each individual component.**

**Rating:** 6
**Confidence:** 3

**Review:**

This paper studies an important area in RL, hierarchical RL, which improves data efficiency by incorporating abstractions. In this paper, the authors proposes an efficient option learning algorithm, which utilizes a TD(0) type objective and constrains the learned policy being not too far away from the past policy. In terms of different abstractions, the paper studies action abstraction through a mixture policy, and temporal abstraction through explicitly limiting the maximum number of switches between options.

The method is well-motived in general, however, I feel the notation in Alg 1 is a little bit unclear, what is \pi' and Q'? The ablation study is well-done, to separate the effects of different types, and gives practitioners some useful guidelines. There is one thing I am curious about, do you try the methods using some online data? Since the paper argues the improvement compared with online option learning, it would be great to also have some experiments using online data for a fair comparison.

I am not that familiar with hierarchical RL, so I could not give a fair judgement of the novelty compared with previous option learning literature. In terms of quality, it is a well-motivated work, clearly written in most part of the paper and gives a method with reasonably good empirical performance. I feel the off-policy argument in this paper is less clear, is it just achieved by using a Q-learning based method? This can also be used online, and how is the online-version compared with the actor-critic option learning method?

---

> ### Author Response · Authors · 2020-11-17
> **Individual feedback per review**
>
> We thank the reviewer for their detailed and constructive feedback. In this post, we will focus on reviewer specific feedback. We will additionally provide an overview post describing the general feedback and corresponding changes.
>
> - Online/offline versus on-policy/off-policy learning
>
> It is unclear if we correctly understand the comment regarding the use of online data as off-policy methods in general still use online data and will try to clarify in the following paragraphs.
> In general, this paper only focuses on the online reinforcement learning setting, and we do not address the offline RL problem (where the policy must be learned entirely from existing data, without interaction). However, HO2 is an off-policy reinforcement learning algorithm, meaning that it can learn from interactions that do not (only) come from the current policy itself.
>
> The experiments show that (a) common existing approaches to hierarchical RL (which are on-policy) can be outperformed by strong non-hierarchical off-policy methods like MPO, thus showing the criticality of learning off-policy; and (b) our proposed off-policy hierarchical approach outperforms other methods with the same underlying optimization algorithm, such as RHPO and MPO, showing the benefit of a hierarchical off-policy method. We have added more discussion to clarify some of these nuances.
>
> - Unclear notation in Algorithm 1 and what are \pi' and Q'
>
> We addressed the following points directly in the paper. \pi' and Q' respectively represent the target policy and target Q-function as previously indicated via the corresponding comment in the algorithm. We have clarified this aspect and provided additional information for other algorithm parameters.
>
> If not addressed explicitly in the author feedback, we will address the remaining minor comments directly in the paper. Please do not hesitate to emphasise remaining questions or concerns.

---

### Official Review · AnonReviewer2 · 2020-10-29
**Official Blind Review #2**

**Rating:** 3
**Confidence:** 3

**Review:**

## Summary
This paper introduces a novel option-learning policy gradient method, HO2. The method learns a parameterized joint distribution over options and actions and uses a soft-continuation based approach to interrupt or "switch" between options before option termination. The method introduces a new meta-parameter which enforces a hard limit on the number of "switches" that can occur, significantly reducing the variance of the option-learning method and replacing softer loss penalization based approaches. The paper demonstrates the performance of the proposed algorithm on a handful of 3D virtualized environments as well as on robotic simulation tasks.

## Review
### Summary
I am currently leaning towards recommending reject for this paper. While the approach and algorithm are novel, they also appear to be highly complex, don't provide a noticeable or consistent improvement over much simpler benchmarks, and I fear that the improvements that _are_ seen are likely due to variance in the results or are hidden behind the additional machinery in the algorithm. I remain slightly skeptical of the utility of the proposed meta-parameter $n$ for setting a hard limit on the number of "switches" between active options, with my skepticism primarily due to concerns on the difficulty of tuning this parameter and the domain-specificity of the parameter.

### Details
I'm curious about some of the hidden complexities in the algorithm. In the problem setup, a policy is defined on an MDP as possibly being $\pi(a | h)$ where $h=\{s_t, a_{t-1}, s_{t-1}, a_{t-2}, \ldots, s_0\}$,  that is a full history of interactions. First, I suppose this means we are no longer dealing with the original MDP and are working in a modified MDP where the "Markov" state is a full history; already leading towards an exponential growth of the state-space. The algorithm itself depends on a recursive product of distributions for the entire length of a sampled trajectory as a result of this adapted Markov state. I'm initially worried at how difficult it is to keep this probability from decaying towards 0 rapidly. There appear to be multiple partial marginalizations (e.g. $\sum_{i = 0}^M p(X | Y = y_i) p(Y=y_i)$) which require rescaling the final product to stay within the standard simplex, perhaps this is used to prevent the product of distributions from decaying towards 0?

One of the primary motivations of the hard switching limit is that an auxiliary penalization on the objective is hard to tune. However, it isn't clear to me that the hard limit parameter $n$ would be any easier to tune. In fact, because the inclusion of $n$ seems to require more partial marginalizations, it almost seems as if this would cause additional complexity in the optimization problem. Did you find this meta-parameter easy to tune? What are the effects of choosing it to be $n=5$ for the experiments instead of (say) 10? It appears to play a very mild variance reduction role in the results (though with only 5 seeds, we _really_ can't say much about variance since this is severely underestimating variance). If the switching limit prefers to stay small, would this suggest that the best form of the algorithm is one without the options framework at all (e.g. the best version of the proposed algorithm is an unaltered actor-critic algorithm)?

The experiments in the paper start with deep neural networks with all of the necessary machinery to make Deep RL run at the moment, including experience replay, target networks, ADAM optimizer, layer norms, mini-batches, various types of activations on each layer, neural networks of different architectures for each of the three sets of weights, different stepsizes for each of the networks, etc. I fail to see why this approach couldn't have been studied in a much simpler linear function approximation setting where statistically significant results with fewer confounding variables could have been achieved. As it stands, it is entirely unclear to me if the proposed algorithm actually provides any benefit when, in the midst of all of the machinery, the modifications above the benchmark algorithms are modest. Given this, there certainly is something to be said for a novel algorithm that does perform favorably when included in the machinery of a Deep RL feat of engineering.

I , however, am unsure if the proposed algorithm does perform favorably. From the results, it appears that in most cases RHPO performs equivalently to the proposed; certainly not statistically significantly different. With only 5 random seeds, it would be very hard to make sound claims; especially considering the known variance issues in Deep RL (take Henderson et al. 2018 for a deeper discussion). The one place where the proposed algorithm _does_ outperform RHPO is in the robot simulator (though again with only 5 seeds, heavy skepticism is called for). I found this fascinating and am curious if there is some structure that the proposed algorithm is able to take advantage of in this domain that RHPO is unable to replicate.

# After discussion period:
I have read all other reviews, resulting conversation, and have read the edits to the paper. After extended conversation with the authors and a deeper investigation into the empirical components of the paper, I find I have further concerns than originally realized in my original review and that many of my original concerns remain. I am lowering my recommendation from a 5 -> 3 to reflect the new concerns; namely the validity of the ablation study as detailed in-depth below.

---

> ### Author Response · Authors · 2020-11-17
> **Individual feedback per review**
>
> We thank the reviewer for their detailed and constructive feedback. In this post, we will focus on reviewer specific feedback. We will additionally provide an overview post describing the general feedback and corresponding changes.
>
> - Working with an extended MDP
>
> Yes, it is correct that the options framework in general (not just our method) extends the MDP. Our paper uses the usual semi-MDP model [3] which is common across option frameworks. Although our notation in Equation 3 remains general in that the policy at time t is dependent on the entire history, this dependence is later refined to the kind of dependence that is described in the semi-MDP framework (i.e. the dependence is mediated by the option which is described in Equation 4). We do not discuss semi-MDP framework in detail in the paper, but there are relevant discussions to be found for example in [1, 2, 3]. In addition, we have clarified this aspect in the paper.
>
> - Details for the switch constrained extension
>
> The problem of controlling when to switch between options is a known challenge (e.g. [1]). For most domains, it is usually not known a priori what would be appropriate switching behaviour. The possibility to explicitly optimise for temporal consistency via the switch constraints, which can further improve performance, is an additional feature of HO2 (as shown in Figure 6).
> Our presented approach aims to simplify tuning this aspect. It is easier to tune than another weighted cost term (as e.g. given in [1]) since it can be chosen independently of the reward scale and does not directly cause a potential conflict with other reward terms. In addition, setting how many times the agent should maximally switch between options along a trajectory can be more intuitive than an additional cost term, which is missing a similarly easy semantic interpretation. We will emphasise this further in the paper.
>
> If the switching limit prefers to stay small, would this suggest that the best form of the algorithm is one without the options framework at all (e.g. the best version of the proposed algorithm is an unaltered actor-critic algorithm)?
> The algorithm often converges to a low rate of switching but never (in our experiments) degrades to the single option case (which would be equivalent to a non-hierarchical policy). This shows that even when the agent would be able to change its behaviour to represent the ‘unaltered actor-critic algorithm’, there are benefits in using multiple options. Further, note that MPO trains a flat single actor-critic model, and yields poorer performance. We have clarified this in the experiments section.
>
> - Simpler setting e.g. linear function approximation setting
>
> Simplifying evaluation to obtain more general insights is an important point. However, designing toy domains that still include the relevant aspects which render the real-world problems hard is challenging on its own.
> We investigate option learning in particular in the context of deep models as these can represent solutions to more complex tasks which share more aspects with real-world control problems. While sharing some of the challenges with deep models, the optimisation of linear models is commonly affected by different aspects of the optimization problem.
>
> While clearly multiple factors affect performance (as the reviewer suggests), we carefully ablate over these in the experiments to identify their effect. These experiments include:
> 1) Comparison to current on-policy option algorithms to estimate the importance of off-policy learning in HO2.
> 2) Comparison to flat and mixture policies (MPO and RHPO) with equivalent underlying policy optimization to independently evaluate the benefits of temporal and action abstraction.
> 3) Ablation over the use of switching constraints, action-conditioning, off-policyness, and robustness via trust region constraints.
> 4) Analysis and interpretation of option decompositions for simpler tasks.
>
> - Performance differences on the simpler benchmarks and in general
>
> We have changed the paper to be more specific with respect to domains where HO2 provides the strongest performance gains. However, across all domains, the proposed method performs either better than existing methods or at least on par.
>
> - Variance of results and number of seeds
>
> We agree that variance and significance of results in reinforcement represent a critical point. Commonly, authors have to trade-off between increasing accuracy of estimates and acceptable computational cost. We run 5 seeds per algorithm per task (a number common across RL papers and also used in [4] for most experiments). We see consistent results across many tasks with stronger benefits dominant in more complex tasks.
>
>
> [Feedback continues in the next post]

---

> > ### Author Response · Authors · 2020-11-17
> > **Continuation**
> >
> > - Structure that HO2 is able to take advantage of that RHPO is unable to replicate
> >
> > The core difference between HO2 and RHPO is the ability to represent temporal abstraction with respect to the options (i.e. the ability to explicitly model the continuation of the behaviour of an option instead of resampling at every timestep). This requires the presented changes in the Q-function and the modelling of high-level option probabilities via dynamic programming. As pointed out by the reviewer the aspect seems to be particularly beneficial in the visually complex 3D manipulation tasks. This suggests that the additional structure for exploration is particularly helpful here. We have further clarified this in the paper.
> >
> > If not addressed explicitly in the author feedback, we will address the remaining minor comments directly in the paper. Please do not hesitate to emphasise remaining questions or concerns.
> >
> >
> > [1] Levy, K. Y. and Shimkin, N. (2011). Unified inter and intra options learning using policy gradient methods. In Proceedings of the 2011 European Workshop on Reinforcement Learning.
> >
> > [2] Puterman, M. L. (2014). Markov decision processes: discrete stochastic dynamic programming. John Wiley & Sons.
> >
> > [3] Sutton, R. S., Precup, D., & Singh, S. (1999). Between MDPs and semi-MDPs: A framework for temporal abstraction in reinforcement learning. Artificial intelligence, 112(1-2), 181-211.
> >
> > [4] Henderson, P., Islam, R., Bachman, P., Pineau, J., Precup, D., & Meger, D. (2017). Deep reinforcement learning that matters. arXiv preprint arXiv:1709.06560.

---

> > > ### Comment · AnonReviewer2 · 2020-11-20
> > > **Feedback**
> > >
> > > I appreciate the detailed response!
> > >
> > > > Working with an extended MDP.
> > >
> > > I agree that this history-MDP is not novel and has been studied in the literature, but it is incongruent with the setting that the paper currently states it is solving. I think this could be a bit misleading, especially considering that the solution method in the experiments does not deal with the hidden difficulties inherent with history-MDPs (e.g. there is no recurrence, or historical inputs to the network, etc.). In this way, the paper reads like: "Define problem A, develop theory for problem B, investigate solution on problem A". Reading the paper right now, I might fool myself that this algorithm is searching for a solution in $\mathcal{S} \times \mathcal{A}$-space since it appears to be an MDP problem with that state-space, but in reality the theory in Section 3 only tells me how to find a solution that is in an exponentially larger space $\left( \mathcal{S} \times \mathcal{A} \right)^T$ where $T$ is number of timesteps of the longest possible trajectory. I don't want to sound alarmist and say that this makes the approach hopeless, but I am a little concerned that the paper seems to jump back and forth between an easier-to-solve problem and a scary-big problem very subtly. Concretely, I think the best resolution is to "Define problem B, develop theory for problem B, mention that problem B is hard so investigate if solutions in the sub-problem A are worthwhile".
> > >
> > > I only just noticed, but an additional concern on this front is this statement from the paper: "Note that even though we express the policy as a function of the history $h_t$, $Q$ is a function of $o_t, s_t, a_t$, since these are sufficient to render the future trajectory independent of the past." This is untrue. A value function $Q$ is defined in terms of a policy (note that $Q_\pi$ is the estimate of a return _following policy_ $\pi$). This means the value function is additionally a function of $\pi$ and thus is also a function of history. I have some concern that---held under a microscope---there are a few claims and implicit assumptions made in the paper that might break due to the mismatch between the formal problem statement and the algorithm derivation.
> > >
> > > > Details for the switch constrained extension
> > >
> > > The paper and response both make the claim that setting a hard limit on switches is easier than a soft limit. My experience with setting hard limits pulls me to believe otherwise, so I remain skeptical. I had hoped for some support within the paper investigating this claim. Concretely, if I were to deploy this algorithm in a never-ending (or just long-term deployment) environment, I would be far more likely to know information like the scale of my rewards than the maximum number of times my algorithm should switch from one abstract option to another. In fact, I would likely think that my algorithm should infinitely many such switches and would simply want a penalization term to discourage switching more often.
> > >
> > > Of course, I don't want to impose my opinions on algorithm design here because I am very likely wrong. But I would like to be shown that my gut-reaction is wrong, instead of needing to take the claims on faith. A single sensitivity curve for $n$ would have gone a long way towards supporting this claim I think.
> > >
> > > I do appreciate the note that $n$ is never chosen to be 0. I think these sort of sanity checks cannot be overstated in an empirical section; they demonstrate that the algorithm is at least doing what it claims to be doing (a standard that not all RL papers aspire to dishearteningly).
> > >
> > > [ran out of space, continuing in a follow-up]

---

> > > > ### Comment · AnonReviewer2 · 2020-11-20
> > > > **Feedback Part 2**
> > > >
> > > > > Simpler setting e.g. linear function approximation setting
> > > >
> > > > On one hand, I totally agree that designing small toy problems that represent the real world is hard and is a waste of time, on the other hand I wonder if that is the wrong goal anyway. Rather I would like to see experimentation done in two steps (A) demonstrate the existence of a problem and show that I have a solution (B) show that this seems to help on real-world problems. A beautiful intermediary step would be to show that the problem in A exists in B, but that can be prohibitively hard and some amount of suspension-of-disbelieve is always appropriate for conference papers. This removes the burden of making A simulate the real-world and instead focuses on distilling exactly what problem the paper is proposing a solution to, then showing me that the paper solves that problem. A nice side-effect being that the clarity of the paper increases by an order of magnitude from this demonstration. Further, because toy problems are often cheap to run, statistical significance is no longer a tradeoff between compute and ability to support claims.
> > > >
> > > > By skipping straight to the real-world domains, it isn't immediately clear that the proposed algorithm actually solves a problem. Right now, a reasonable explanation of the results that I see in Section 4 is that we rolled the proverbial experiment dice a few times and came out slightly ahead. But if first the reader is convinced that the proposed algorithm solves a concrete problem, and the reader is willing to believe that this problem exists in the real world, then now the 5 random seeds and lack of statistical significance tells a stronger story because it builds on a pre-existing structure and expectation.
> > > >
> > > > > Variance of results and number of seeds
> > > >
> > > > Taking some time to consider this further and I partially concede your point. Because there are so many experiments each with 5 random seeds (and at least most of these are definitely independent) and because the proposed algorithm outperforms or ties its competitor in nearly all of these additional domains, then we can assume that this demonstrates a superiority of the proposed algorithm over its benchmarks. However, there still is not enough evidence to support any single claim with the experiments due to the variance and number of seeds. So the depth of the ablation study is still entirely unconvincing for its purpose because of this.
> > > >
> > > > So in summary, I agree that there is likely enough evidence to claim that HO2 outperforms RHPO and MPO on average, though I do not think the degree to which it outperforms is substantial. I still do not believe any of the ablation studies provide any additional evidence that HO2 outperforms competitors when randomizing over particular factors. I believe this is yet to be adequately supported empirically.
> > > >
> > > > ## Summary after author response
> > > > In light of the above, I so far intend to keep my recommendation as is. My primary reasons are (A) inconsistency across formal problem settings (B) no falsifiable support for the new meta-parameter and not strong intuitive support either (C) insufficient empirical support for claims in paper, with the exception of the implicit claim that HO2 outperforms competitors where I still find the degree of improvement may not outweigh the increased complexity of the proposed algorithm.

---

> > > > > ### Author Response · Authors · 2020-11-24
> > > > > **Feedback**
> > > > >
> > > > > Thank you very much for the additional feedback. We are pleased that the last rebuttal clarified some questions but see that there are still remaining misunderstandings and will try to clarify these in the following paragraphs.
> > > > >
> > > > > - Extended MDP
> > > > >
> > > > > To start with the most important point: the option policy does only rely on the history h via the option o, and does not require explicit dependence on the entire history. This is described in the equations and graphical model but we will again strengthen this point in the paper.
> > > > >
> > > > > Therefore, the setting in this paper is identical to most previous work on option learning [1, 2, 3, 4, 5]: the semi-MDP. At each timestep, the option policy is only dependent on the current state and the previous option, as described in Equation 2, and the joint action-option probability can be decomposed following Equation 3. We have further clarified that the history-based notation was chosen to describe the connection to mixture policies.
> > > > > Since the policy explicitly only depends on the option (and not the full history), the same also holds for the state-action value function Q which is now a function of state, action and option Q(s,a,o). We hope this fully addresses the second concern.
> > > > >
> > > > > In addition to previous changes, we have now clarified this aspect in the method section and also more clearly referred to the semi-MDP framework.
> > > > >
> > > > > - Details for the switch constrained extension
> > > > >
> > > > > We would like to start the answer for this point by emphasising that the core method does not require switch constraints and this is purely an extension (made possible by the structure of the inference graph).
> > > > >
> > > > > In addition, we would like to clarify the argument on reward scale independence of this hyperparameter, which provides one of the big improvements over commonly used additional weighted costs. Essentially, this hyperparameter can be set independent of an environment’s reward scale since the method does not directly affect the objective which is being optimised but rather the paths through the inference graph which are used for achieving the objective.
> > > > >
> > > > > - Simpler setting
> > > > >
> > > > > We agree on the importance of toy domains and use a domain which is common across related literature with the OpenAI gym experiments. However, we expect that the reviewer aims at even simpler domains which could help to further investigate different algorithm aspects.
> > > > >
> > > > > In general, the recent option learning literature purely focuses on domains of the complexity of our simple OpenAI gym domains - in many cases, papers only use these exact domains (e.g. [4, 5]) . While we do not extend the common set of experiments to even simpler domains, we move in the opposite direction and improve the analysis in our experimental section by extending to more realistic domains and complex raw pixel inputs.
> > > > > In this way, our experiments are already of increased detail in comparison to recent publications. We have chosen to focus on scalability and robustness for this work because these domains are closer to a practical, real-world application.
> > > > >
> > > > > - Variance and number of seeds
> > > > >
> > > > > We appreciate the reviewer recognising that HO2 outperforms baselines over multiple seeds and over many experiments. However, we would like to address the comments regarding that they “do not think the degree to which it outperforms is substantial”, and “do not believe any of the ablation studies provide any additional evidence that HO2 outperforms competitors when randomizing over particular factors.”
> > > > > Our main concern is that these comments remain ambiguous and do not point to specific issues which we as authors can address but rather a general opinion of the reviewer. Without any specific missing experiments or missing ablations this renders it impossible to address, and in our opinion, fails to carefully consider the various ablations already performed in the paper.
> > > > >
> > > > > In addition, the use of 5 seeds for experiments is common across RL publications and was in addition applied for most experiments in the paper previously cited by the reviewer ([6] and other references).
> > > > >
> > > > > We would welcome actionable feedback regarding these points, and indeed have already taken previous comments to further improve the paper. However, as discussed in our previous response, we already have designed the experiments in the paper to rigorously measure the contribution of HO2 and ablate the impact of different components.

---

> > > > > > ### Author Response · Authors · 2020-11-24
> > > > > > **References for feedback**
> > > > > >
> > > > > > [1] Levy, K. Y. and Shimkin, N. (2011). Unified inter and intra options learning using policy gradient methods. In Proceedings of the 2011 European Workshop on Reinforcement Learning.
> > > > > >
> > > > > > [2] Puterman, M. L. (2014). Markov decision processes: discrete stochastic dynamic programming. John Wiley & Sons.
> > > > > >
> > > > > > [3] Sutton, R. S., Precup, D., & Singh, S. (1999). Between MDPs and semi-MDPs: A framework for temporal abstraction in reinforcement learning. Artificial intelligence, 112(1-2), 181-211.
> > > > > >
> > > > > > [4] Zhang, S., & Whiteson, S. (2019). DAC: The double actor-critic architecture for learning options. In Advances in Neural Information Processing Systems (pp. 2012-2022).
> > > > > >
> > > > > > [5] Smith, M., Hoof, H., & Pineau, J. (2018, July). An inference-based policy gradient method for learning options. In International Conference on Machine Learning (pp. 4703-4712).
> > > > > >
> > > > > > [6] Henderson, P., Islam, R., Bachman, P., Pineau, J., Precup, D., & Meger, D. (2017). Deep reinforcement learning that matters. arXiv preprint arXiv:1709.06560.

---

> > > > > > ### Comment · AnonReviewer2 · 2020-11-24
> > > > > > **Feedback**
> > > > > >
> > > > > > The author response raises concerns of ambiguity in my feedback and seeks direct, actionable feedback. I will try to be even more direct here to reduce any potential future miscommunication. Unfortunately, directness tends to read as bluntness or even rudeness, so please note that the negativity (a) is biased due to the conversation being focused around what I perceive as being detriments of the paper and (b) is likely overstated in the attempts to be more clear. I will make sure my final recommendation of the paper reflects this.
> > > > > >
> > > > > > > Extended MDP
> > > > > >
> > > > > > Section 2 defines an MDP with statespace $\mathcal{S}$, but with a policy $\pi : \mathcal{H} \times \mathcal{A} \to [0, 1]$. These two statements are incongruent. I appreciate further context for why you care about histories, but this was already clear to me. I am less concerned with why and more concerned with if it is correct.  **Action item:** clarify in Section 2 what setting you are in. Perhaps this can be as simple as citing the Sutton et al. 2009 paper in Section 2 and using their formalism.
> > > > > >
> > > > > > In Equation 3, your parameterized policy $\pi_\theta (a_t, o_t | h_t)$ is a function of history. Equation 6 suggests that $Q_\phi$  depends on $\pi_\theta$ and $q(a_t, o_t | h_t)$. Algorithm 1 (HO2) says component probabilities are determined via $\pi^H (o_t | h_t)$ and actions are sampled from $\pi_\theta(\cdot | h_t)$.  I'll admit, the abundance of symbols and the time between my careful reading of the paper and now makes it difficult for me to interpret what is happening here. However, every single policy that the optimization for $Q_\phi$ seems to depend on, itself depends on history $h_t$. The author response says this is untrue, that $Q_\phi$ does not depend on $h_t$. **Action item:** clarify throughout Section 3 the dependence between $Q_\phi$ and $h_t$. The current stated sentence is insufficient because it is not self-evident. If the graphical model says that $\pi$ is dependent only on $o_t$, then change the notation throughout Section 3 to state that instead of the more general $h_t$.
> > > > > >
> > > > > > > Switch constrained extension
> > > > > >
> > > > > > The clarification about the reward independence is not necessary, that part is clear to me. Admittedly, I am unsure how to make that more clear than my previous response, but I will try. My complaint is the following. The paper and author responses make a claim: a hard limit is easier to use than an additional term on the loss because the additional term on the loss depends on knowing the scale of rewards. However, there is no support for this claim in either the author response or the paper. As such, I must fall back on my own intuitions---which are fallible---and that makes me uncomfortable. My intuition says that I am far more likely to know the reward scale because I design the rewards myself for most real-world problems. I am highly unlikely to know how many times my algorithm should switch options though. I can use a priori information to set a reward scale, but it seems difficult to use prior information to set this new parameter. **Action item:** provide support for the claim that the new extension is easier to use than previous work. This could be done (for instance) by showing that the algorithm is insensitive to the new parameter. This could also be done by describing a real scenario where it is "easy" to know what the hard limit parameter should be, but where the practitioner won't likely know the reward scale.
> > > > > >
> > > > > > > Simpler setting
> > > > > >
> > > > > > I regret that I was unclear in my last response. I should have stated that this is in combination with the random seeds/empirical study complaint; these two comments should not be treated separately. As a whole, the complaint is that there is insufficient empirical evidence to support most claims in the paper. A possible solution (read: an **actionable item**) to avoid computational cost would be to use simpler, better thought-out settings.

---

> > > > > > ### Comment · AnonReviewer2 · 2020-11-25
> > > > > > **Feedback Part 2**
> > > > > >
> > > > > > > Empirical evidence
> > > > > >
> > > > > > Let's have some nuance. Is your claim: "HO2 outperform baselines over multiple seeds and many experiments" true? First, it's impossible to know if it outperforms over multiple seeds since we are only shown an aggregate over seeds (maybe it only outperforms once, and fails four times, but on average still outperforms), so this claim certainly cannot be supported with evidence found in the paper. Second, let's tally the results:
> > > > > > * Figure 3: HO2 outperforms competitors convincingly in 1 of 4 plots by a margin of 20%, ties in other 3.
> > > > > > * Figure 5: HO2 outperforms baselines in 3 of 4 plots by a margin of [10%, 20%, and 15%] roughly judging by final performance . This is your strongest evidence by far.
> > > > > >
> > > > > > In total, we have HO2 winning in 4 of 8 plots with a rough average of 15% improvement where it wins. I can equally conclude that HO2 "outperforms baselines [...] over many experiments" as I can conclude that HO2 does **not** outperform baselines over many domains. Further, on average across all domains including where HO2 ties, we expect an 8% improvement over competitors assuming that all 8 domains are totally independent (share no common features). Of course, this is a bad assumption since in Figure 5 all of the environments are quite similar to each other (different tasks on the otherwise same MDP), implying that 8% is an *overestimate*.
> > > > > >
> > > > > > By performing this aggregating tally, I was willing to concede that there was some evidence that HO2 improves over baselines (I even ignored the fact that this is an overestimate). But that was using 8 plots for a total of 40 random seeds. Let's consider each ablation individually now to understand my point about not being able to make claims about the ablations.
> > > > > >
> > > > > > Figure 6 asks whether the hard-limit plays a role on performance, and how it impacts the option switching rate. The left figure says: no impact. The right figure says: big impact. In total, I couldn't tell you if there is no impact or a big impact. This ablation tells me very little. I don't find any other related figures in the appendix for this ablation. **Action item:** include more domains or include more random seeds to see if the left figure actually has an impact that is hidden behind the variance or try this test on a smaller set of domains where more environments/seeds could be tested.
> > > > > >
> > > > > > Figure 7 asks if conditioning on past actions plays a role on performance. The left figure says: don't condition, the other three say: no impact. In total, I don't know if there is an impact or not. Checking the appendix, and this strongly suggests there is no impact so this wasn't an interesting question. Not only this, but Figure 14 in the appendix suggests that the only appreciable impact was due to plotting choices, not due to actual underlying processes. Further Figure 7 is entirely dishonest in its implications and cannot be called an ablation. It does not control for all aspects between the two presented algorithms, other than the singularly manipulated variable. Instead it additionally: (1) cuts the size of the replay buffer of the poor performing line by 2 orders of magnitude, and (2) it changes the number of learning steps and data collection steps by 1 order of magnitude. **Action item:** conclude that there is no impact and relegate this to the appendix and make the comparison between algorithms more fair (I recognize the attempt to make this fair in Figure 14, but should note that this made the two algorithms perform almost identically).
> > > > > >
> > > > > > Figure 8 asks of the impact of the trust-region constraints. In its singular domain presentation, it appears there are massive impacts on performance. But checking Figure 15 in the appendix, it is clear that this is not consistently true and is only true in the cherry-picked domain presented in the main paper and it's most related sibling. In all other cases, the results are washed away by variance or are nearly identical. **Action item:** relegate this to appendix, or provide a more fair view across domains in main paper. Don't cherry-pick.
> > > > > >
> > > > > > I will point out some interesting details of the Henderson et al. paper. Tables 1 and 2 use 5 random seeds based on code from other works to demonstrate the inconsistency of results. They go on to further demonstrate this in Figure 5 where Henderson et al. uses 5 random seeds to demonstrate the insufficiency of 5 random seeds. The remainder of the plots from Henderson et al. use bootstrap confidence intervals with 10k bootstrap iterations; a far cry from using only 5 random seeds.
> > > > > >
> > > > > > Edit: I realized that the change in formatting from writing the response to posting it resulted in the bolded words being more aggressive than intended. I reworded and reformatted the paragraph about Henderson et al.'s work to make the language less blunt but still direct.

---

### Official Review · AnonReviewer4 · 2020-11-09
**Well-motivated paper that presents a new algorithm for hierarchical reinforcement learning using the options framework. Empirical results demonstrate gains in data efficiency in a number of problems but do not provide substantial insight into the behaviour of the algorithm.**

**Rating:** 5
**Confidence:** 3

**Review:**

The paper introduces a reinforcement learning algorithm with temporal abstraction using the options framework. It provides empirical results in a variety of domains, demonstrating that the algorithm can improve data efficiency.

The paper is well motivated. Data efficiency is an important concern in applications of reinforcement learning.  The approach is sufficiently novel. The empirical results are positive, showing performance improvements in a variety of domains. Results in simulated robotic manipulation tasks are particularly positive, as measured by average return.

The paper can be improved by providing additional analysis to better understand the behaviour of the algorithm and the conditions under which it improves performance. For instance, it would be useful to see what type of options are being learned in the various domains and with different limits on the number of switches allowed.

I had difficulty in evaluating the importance of the performance differences in Figure 5. In the main text of the paper, there is no information on the reward structure of the task. Additional information is present in the  appendix but I could not easily locate the relevant information (if it is indeed there). For instance, it would be useful to know what the behavioural difference is between average returns of 60 and 100.

In figure 3, the number of switches are listed as 5. Some experimentation with various different values would be informative. In figure 5, seeing a longer training period would be informative. MPO and RHPO are still improving at the end of the learning curve.

Please explain Equation 4 in some detail.  I could not follow. In particular,  I do not follow why the  $\pi^{L}$ term is there.

The paper is not easy to read. Many sentences do not communicate the intended meaning clearly. As an example, the first paragraph of the introduction would be difficult to understand by readers who are not already familiar with hierarchical reinforcement learning, the options framework, and the papers cited. And some of the writing is not clear regardless of the background of the reader. For instance, the first paragraph ends with "Overall, the interaction of algorithm and environment can become increasingly difficult, especially in an off-policy setting (Precup et al., 2006)." It is not clear what is meant by a "difficult" interaction here.

The writing could be more nuanced in the discussion of results presented in Figure 3. The authors write that “off-policy learning alone [..] improves data-efficiency and can suffice to outperform on-policy option algorithms such as DAC, IOPG and Option-Critic.” This is not true in every domain. Similarly, the authors write that they “achieve improvements when training mixture policies via RHPO”. Again, this is not true in all four domains. For instance, in Hopper-v2, RHPO lags behind MPO.

In the pseudo code for Algorithm 1 on page 5, please specify inputs to the algorithm. And please do not use any undefined symbols (e.g., $\pi’$, $Q’$).

There is no reference to Figure 6 in the text.

In Figure 3, please include DAC, OC, and IOPG in the figure legend.

In Figure 3, the way the performance of DAC, OC, and IOPG are presented on the plots is misleading. For each algorithm, a constant value is shown from step 0 until step $2 \times 10^6$ although these constants correspond only to average return obtained after $2 \times 10^6$ steps. Furthermore, my understanding is that these numbers have been taken from Zhang & Whiteson (2019). For best experimental  practice, these algorithms should be tested by the authors themselves along with the other algorithms shown in the plot (e.g., HO2). This would ensure that the performances are truly comparable and that there are differences in relevant experimental settings. In addition, it would allow the reader to compare the algorithms along the entirety of their learning curves.

Doina Precup's dissertation is listed twice in the references, with different publications years.

Misuse of the comma is prevalent throughout the paper.

The author response answered some of my questions. But I cannot say that I now better understand the behaviour of the algorithm and the conditions under which it improves performance. I agree with reviewer 2 that analysing behaviour and performance in a simpler setting would be informative. While the writing has improved, it stills lacks the clarity and nuance one would wish to see in a paper at this conference.

---

> ### Author Response · Authors · 2020-11-17
> **Individual feedback per review**
>
> We thank the reviewer for their detailed and constructive feedback. In this post, we will focus on reviewer specific feedback. We will additionally provide an overview post describing the general feedback and corresponding changes.
>
> - Additional analysis of the behaviour of the algorithm; conditions under which it improves performance; types of emerging options
>
> We agree that these aspects are important to understand. We have clarified and extended existing analysis sections. Section 4.2 focuses on differences to a simple Gaussian policy and a mixture of Gaussians policy (which can be understood as a single-step options model). In these sections, we analyse the impact of action abstraction (the ability to control by choosing from a set of individual skills/options) and temporal abstraction (the ability to model consistent behaviour enabled via the termination conditions). Furthermore, Section 4.3 includes the investigation of the impact of off-policy training, trust-region constraints and different aspects affecting the types of option decompositions. For improved understanding, we will include further details in the main paper regarding the properties of learned options based on conditions such as switch constraints and information asymmetry (information only provided as input to a part of the agent) between high- and low-level controller.
>
> - Clarity of writing
>
> Thanks for pointing this out. We have gone through the paper carefully and changed any sentences that may have been unclear. In particular, we adapted the introduction to render the paper more accessible and improve overall readability. Please feel free to use the openreview diff function to trace these changes.
>
> - Gym experiments: On-policy and off-policy option learning; comparing mixture and option policies
>
> In general, off-policy algorithms have a considerable advantage over on-policy methods as they enable multiple updates for the same data samples. This advantage grows when training with stochastic gradient descent and only small adjustments per update. With respect to our experiments, we have explicitly clarified on which domains off-policy learning (including non-hierarchical policies) outperformed existing work on on-policy option learning. Additionally, we have emphasised the equivalent details in the RHPO and HO2 comparison.
>
> - Gym experiments: results from DAC, OC, IOPG
>
> We agree that this is an important figure and there is no perfect way of doing this comparison. We've attempted to do it in a fair manner and will clarify the motivation behind this type of comparison below.
>
> As described in the caption, we use previously obtained results in the gym domains from prior work and have additionally contacted the authors to ensure that the environments and experiment setting are exactly the same. We chose to follow this direction to ensure that we take the best known results in this benchmark (a common practice in many fields e.g. computer vision, NLP and partially used in RL as well), instead of using a sub-optimal reimplementation of the algorithms which would potentially underperform existing results.
>
> Using straight lines to indicate final results after the maximum training time of 2x10^6 steps instead of complete learning curves has two reasons. First, to prevent additional clutter in the graphs. Second, the learning curve comparison between on-policy and off-policy learning is only meaningful within limits. While we can align the number of actor steps, we cannot do so for learner steps as the ratio can be independently chosen in off-policy learning. We will further emphasise the description of the lines in the caption (which we use instead of an additional legend) to ensure the reader is aware of this setting.
>
> - Details for the switch constrained extension
>
> The problem of controlling when to switch between options is a known challenge (e.g. [1]). For most domains, it is usually not known a priori what would be appropriate switching behaviour. The possibility to explicitly optimise for temporal consistency via the switch constraints, which can further improve performance, is an additional feature of HO2 (as shown in Figure 6).
> Our presented approach aims to simplify tuning this property. It is easier to tune than another weighted cost term (as e.g. given in [1]) since it can be chosen independently of the reward scale and does not directly cause a potential conflict with other reward terms. In addition, setting how many times the agent should maximally switch between options along a trajectory can be more intuitive than an additional cost term, which is missing a similar semantic interpretation. Empirically, we have found performance in the learning from scratch results particularly robust with respect to different values of the constraint. We have emphasised this further in the paper.
>
> [Feedback continues in the next post]

---

> > ### Author Response · Authors · 2020-11-17
> > **Continuation**
> >
> > - Robot experiments: difficulty in evaluating the performance differences and longer learning curve
> >
> > To better understand the differences in performance, one can look at the maximum rewards in a task and the number of steps per episode. Details for these aspects are given in the appendix and we will also improve the description to provide additional intuition in the paper.
> > All four tasks displayed in Section 4.2 use sparse binary rewards, such that the obtained reward represents the number of timesteps where the corresponding condition - such as the ball is in the cup - is fulfilled.
> > Even with the given duration, the most important points are present in Figure 5. The learning curve clearly shows the individual improvements in data-efficiency from the introduction of a mixture policy and the extension to include temporal abstraction via the option policy. Both improve performance throughout most tasks.
> >
> > - Equation 4 in detail and the reason for low-level policy term
> >
> > This directly follows from an application of Bayes rule. The equation computes the probability of the option o_t at time t given past states and actions. We compute it by marginalizing out the previous option. Now, since o_{t-1} affects a_{t-1}, we need to take the previous action that was actually chosen into account when performing the marginalization. In other words, we treat the options as hidden variables and use states and actions as observed variables, both of which hold information about the sequence of options. We have added the above brief intuition to the paper.
> >
> > If not addressed explicitly in the author feedback, we will address the remaining minor comments (e.g. duplication of references, additional information for the algorithm) directly in the paper. Please do not hesitate to emphasise remaining questions or concerns.
> >
> > [1] Harb, J., Bacon, P. L., Klissarov, M., & Precup, D. (2017). When waiting is not an option: Learning options with a deliberation cost. arXiv preprint arXiv:1709.04571.

---

### Author Response · Authors · 2020-11-17
**General feedback and changes**

We thank the reviewers for their detailed and constructive feedback. Below we comment on common points and highlight some changes that we have made to the manuscript in response.

While most reviews appreciated the perspective, analysis and performance improvements in the submission, multiple reviewers suggested changes to improve the clarity of the manuscript and asked for additional analyses of the learned options as well as for further details regarding the constrained version of the algorithm.

To improve clarity we have made use of the additional ninth page. This has allowed us to include many details which had to be omitted from the submission to remain within the page limit.
In particular, we have expanded the general discussion of option models and included additional details regarding prior work which will hopefully make the paper more self-contained. We have also expanded the technical description in the methods section, as well the analyses of the results.

One focus of our paper is to analyze the impact of different algorithmic aspects on performance and type of learned behaviour. The existing evaluations include the comparison of the impact of action abstraction (via mixture policies) and temporal abstraction (via option policies), the additional optimisation to increase temporal abstraction (via the switch constraints), the impact of trust-region constraints, off-policy learning and finally information asymmetry. To account for the additional requests in the reviews, we extended the corresponding sections and in particular added results regarding the behaviour decomposition via options in Section 4.3.

The possibility to explicitly optimise for temporal consistency via the switch constraints, which can further improve performance, is an additional feature of HO2 (as shown in Figure 6).  Optimising temporal consistency can generally be challenging. Our presented approach aims to at least partially simplify this step. It is easier to tune than another weighted cost term (see details in individual responses) since it can be chosen independently of the reward scale and does not directly cause potential conflicts with other reward terms.
In addition, setting how many times the agent should maximally switch between options along a trajectory can be more intuitive than weighting an additional cost term, which is missing a similarly easy semantic interpretation. We have emphasised this further in the paper.

Please find the detailed individual feedback separately posted under each review. We thank the reviewers again for their help in strengthening the paper and hope that all questions have been answered in the individual sections.

---

### Decision · Program_Chairs · 2021-01-07
**Final Decision**

**Decision:**

Reject

**Comment:**

There was a fair amount of discussion about the paper.  Several reviewers felt that the paper would have been stronger if it tried to do less but better.  The reviews describe in detail what the reviewers would have found compelling, but the key suggestion is to remove the complexity that is not essential for the approach to provide consistent improvements.  Doing this requires a better understanding of the algorithm's behavior and a valid ablation study, a new concern raised during the discussion with the authors.

The reviewers felt that the proposed approach is potentially interesting and would like to see this paper done well.